

# Complex refractive index and single scattering albedo of Icelandic dust in the shortwave spectrum

Clarissa Baldo[1], Paola Formenti[2], Claudia Di Biagio[2], Gongda Lu[1,a], Congbo Song[1,b], Mathieu Cazaunau[2], Edouard Pangui[2], Jean-Francois Doussin[2], Pavla Dagsson-Waldhauserova[3,4], Olafur Arnalds[3], David Beddows[1], A. Robert MacKenzie[1], and Zongbo Shi[1]

[1] School of Geography, Earth and Environmental Sciences, University of Birmingham, Birmingham, B15 2TT, United Kingdom
[2] Université Paris Cité and Univ Paris Est Creteil, CNRS, LISA, F-75013 Paris, France
[3] Agricultural University of Iceland, Keldnaholt, Reykjavik, Iceland
[4] Faculty of Environmental Sciences, Czech University of Life Sciences Prague, Prague, Czech Republic
[a] now at: Department of Geography, University College London, London, WC1E 6BT, United Kingdom
[b] now at: National Centre for Atmospheric Science (NCAS), Department of Earth and Environmental Science, University of Manchester, M13 9PL, Manchester, United Kingdom

*Correspondence to*: Clarissa Baldo (c.baldo@bham.ac.uk)

**Abstract** Icelandic dust can impact the radiative budget in high-latitude regions directly by affecting light absorption and scattering and indirectly by changing the surface albedo after dust deposition. This tends to produce a positive radiative forcing. However, the limited knowledge of the spectral optical properties of Icelandic dust prevents an accurate assessment of these radiative effects. Here, the spectral single scattering albedo (SSA) and the complex refractive index ($m = n - ik$) of Icelandic dust from five major emission hotspots were retrieved between 370-950 nm using online measurements of size distribution and spectral absorption ($\beta_{abs}$) and scattering ($\beta_{sca}$) coefficients of particles suspended in a large-scale atmospheric simulation chamber. The SSA($\lambda$) estimated from the measured $\beta_{abs}$ and $\beta_{sca}$ increased from 0.90-0.94 at 370 nm to 0.94-0.96 at 950 nm in Icelandic dust from the different hotspots, which falls within the range of mineral dust from northern Africa and eastern Asian. The spectral complex refractive index was retrieved by minimizing the differences between the measured $\beta_{abs}$ and $\beta_{sca}$ and those computed using the Mie theory for spherical and internally homogeneous particles, using the size distribution data as input. The real part of the complex refractive index ($n(\lambda)$) was found to be 1.60-1.61 in the different samples and independent on wavelength. The imaginary part ($k(\lambda)$) was almost constant with wavelength and was found to be around 0.004 at 370 nm and 0.002-0.003 at 950 nm. The estimated complex refractive index was close to the initial estimates based on the mineralogical composition, also suggesting that the high magnetite content observed in Icelandic dust may contribute to its high absorption capacity in the shortwave spectrum. The $k(\lambda)$ values retrieved for Icelandic dust are at the upper end of the reported range for low-latitude dust (e.g., from the Sahel). Furthermore, Icelandic dust tends to be more absorbing toward the near-infrared. In Icelandic dust, $k(\lambda)$ between 660-950 nm was 2-8 times higher than most of the dust



samples sourced in northern Africa and eastern Asia. This suggests that Icelandic dust may have a stronger positive direct
radiative forcing on climate which has not been accounted for in climate predictions.

# 1 Introduction

Iceland is a major contributor to the dust aerosol loading over high-latitude (>60° N) regions in the Northern Hemisphere
(Arnalds et al., 2016; Baddock et al., 2017; Bullard et al., 2016; Dagsson-Waldhauserova et al., 2019; Groot Zwaaftink et al.,
2017; Groot Zwaaftink et al., 2016; Meinander et al., 2022; Prospero et al., 2012). With an area of more than 45,000 km$^2$, the
Icelandic desert is the largest in the Arctic (Arnalds et al., 2016) and being subject to intense wind erosion, Iceland is one of
the most active dust source areas in the world, producing about 4.6-40 Tg of dust per year (Arnalds et al., 2016; Groot
Zwaaftink et al., 2016) and 34-135 dust events per year (Dagsson-Waldhauserova et al., 2014; Nakashima and Dagsson-
Waldhauserová, 2019). Near glaciers, dust hotspots cover about 500 km$^2$ and are fed by fluvial-glacial processes acting on
volcanic deposits and can produce millions of tons of dust during single dust events (Arnalds et al., 2016).
Icelandic dust can reach several kilometers in altitude (Dagsson-Waldhauserova et al., 2019; Groot Zwaaftink et al., 2017)
and be transported over long distances to the high Arctic (Baddock et al., 2017; Groot Zwaaftink et al., 2017; Groot
Zwaaftink et al., 2016; Moroni et al., 2018; Prospero et al., 2012). Arctic dust events typically occur during summer/autumn
in northeastern Iceland (Dagsson-Waldhauserova et al., 2013; Dagsson-Waldhauserova et al., 2014; Meinander et al., 2022;
Nakashima and Dagsson-Waldhauserová, 2019), while subarctic dust events mainly occur during winter/spring in southern
Iceland (Dagsson-Waldhauserova et al., 2014; Dagsson-Waldhauserova et al., 2015; Dagsson-Waldhauserova et al., 2019).
A modelling study of Groot Zwaaftink et al. (2016) estimated that Iceland accounts for approximatively 2% of the total
atmospheric dust load in the Arctic. According to this study, high-latitude dust contributes to about 27% of the dust load,
while the remaining amount is dominated by low-latitude dust from Africa and Asia. High-latitude dust is confined at low
altitudes due to the stratified atmosphere in cold climates and dominates dust surface concentrations (Bullard, 2017; Groot
Zwaaftink et al., 2016; Shi et al., 2022). Dust concentration and dust load have important consequences for the radiative
balance in the Arctic, where the radiative effect of light-absorbing particles is enhanced by the presence of snow and ice
(Schmale et al., 2021). Icelandic dust absorbs solar – shortwave (SW) radiation (Zubko et al., 2019). Absorbing aerosol
particles in the Arctic upper troposphere tend to cool the surface, whereas those located at lower altitudes tend to warm the
surface (Schmale et al., 2021). The estimated instantaneous radiative forcing (IRF) of Icelandic dust is 0.005 W m$^{-2}$ at the
top of the atmosphere (TOA) via dust-radiation interaction, which is lower but still significant compared to the IRF at the
bottom of the atmosphere (BOA) via dust deposition onto snow-covered areas, 0.02 W m$^{-2}$ (Kylling et al., 2018).
Icelandic dust accounts for around 5.8% of the total dust deposition in the Arctic. Experiments have shown that the
deposition of volcanic dust particles from Iceland onto snow and ice can reduce the surface albedo and influence melting
(Dragosics et al., 2016; Gunnarsson et al., 2021; Meinander et al., 2014; Möller et al., 2016; Möller et al., 2018; Peltoniemi
et al., 2015; Wittmann et al., 2017). Peltoniemi et al. (2015) found that volcanic sand from the Mýrdalssandur hotspot in




southern Iceland reduced the snow albedo in the wavelength (λ) range 400-2500 nm similarly to black carbon (BC). Laboratory measurements conducted on pure volcanic sand (deposited particles) showed a spectral reflectance of less than 0.04, which is close to a black body (0.0) (Dagsson-Waldhauserova, 2014; Peltoniemi et al., 2015).

In addition, Icelandic dust is a source of ice-nucleating particles (e.g., Sanchez-Marroquin et al., 2020; Shi et al., 2022).
Sanchez-Marroquin et al. (2020) estimated that during summer days, Icelandic dust is the dominant source of ice-nucleating dust particles over areas near the eastern coast of Greenland. The increase in ice-nucleating particles can reduce the amount of supercooled liquid water and consequently the albedo of mixed-phase clouds (Vergara-Temprado et al., 2018), which have a large impact on the radiative budget in the Arctic (Morrison et al., 2012; Murray et al., 2021). Icelandic dust also contributes to the deposition of dissolved iron (Fe) to the subpolar North Atlantic Ocean (Arnalds et al., 2014; Baldo et al.,
2020), which has the potential to alter nutrient biogeochemical cycling and primary production in the surface ocean (Achterberg et al., 2013; Achterberg et al., 2018). The deposition of nutrients such as phosphorus (P) and Fe from mineral dust can also promote the blooming of algae in ice, which consequently reduces the ice albedo and accelerates melting (Cook et al., 2020; Lutz et al., 2015; McCutcheon et al., 2021).

The mineralogical composition of Icelandic dust is considerably different from typical low-latitude dust such as northern
African and eastern Asian dust (Baldo et al., 2020), and the absorption of solar radiation by mineral dust is strongly related to the content of total Fe and Fe oxide minerals (Caponi et al., 2017; Derimian et al., 2008; Di Biagio et al., 2019; Engelbrecht et al., 2016; Formenti et al., 2014; Lafon et al., 2006; Moosmuller et al., 2012; Sokolik and Toon, 1999). Icelandic dust primarily consists of amorphous basaltic material, but it is also enriched in Fe contributing to over 10% of the total dust mass, with relatively high magnetite ($Fe_3O_4$) content (Baldo et al., 2020). This suggests that the optical properties
of Icelandic dust may be also different from that of low-latitude dust.

The complex refractive index, represented by m = n - ik, links the particle chemical composition and optical properties, with the real part n defining the scattering properties and the imaginary part k the absorption properties (Bohren and Hufmann, 1998). In a study of Zubko et al. (2019), the index k of suspended particles from the Mýrdalssandur hotspot was found to be 0.01 at 647 nm, which is around one order of magnitude higher than the average values reported for mineral dust from
northern Africa and eastern Asia by Di Biagio et al. (2019). However, the spectral dependence of k is unknown to date.

In Baldo et al. (2020), we studied the chemical and mineralogical composition of Icelandic dust from several major emission hotspots and found that it is distinctive and could be highly light-absorbing. Here, we go a step forward and, for the first time, determine the complex refractive index and single scattering albedo (SSA) of Icelandic dust over a broad range of λ (370-950 nm) based on in situ measurements of the size distribution and optical properties of suspended particles generated
from natural parent soils.



## 2 Methodology

This section illustrates the approach used to retrieve the SSA and complex refractive index of Icelandic dust in the SW range which was based on the method by Di Biagio et al. (2019). SSA is the ratio of the scattering coefficient ($\beta_{sca}$) to the extinction coefficient ($\beta_{ext}$) – the sum of $\beta_{sca}$ and the absorption coefficient ($\beta_{abs}$). The spectral coefficient $\beta_{abs}$ and $\beta_{sca}$, and the size distribution of suspended dust particles were measured using the multi-instrument atmospheric simulation chamber CESAM – French acronym for Experimental Multiphasic Atmospheric Simulation Chamber (Wang et al., 2011). The complex refractive index was retrieved from the comparison between the spectral coefficients $\beta_{abs}$ and $\beta_{sca}$ measured and computed with Mie theory assuming that the particles are spherical and internally homogeneous, using the size distribution data as input. Although mineral dust aerosol is neither spherical nor internally homogeneous (e.g., Kandler et al., 2007; Okada et al., 2001), we have simplified the calculations of the optical properties to allow comparison with previous studies (Deguine et al., 2020; Di Biagio et al., 2019; Kylling et al., 2018; Reed et al., 2018; Zubko et al., 2019). A schematic diagram of the method used to retrieve the complex refractive index is shown in Fig. 1.

### 2.1 Experimental setup

The size distribution and optical properties of Icelandic dust were measured by the use of CESAM, a 4.2 m$^3$ stainless steel atmospheric simulation chamber (Wang et al., 2011). The experiments were conducted using surface sediment samples from five major dust hotspots in Iceland (Baldo et al., 2020) – D3 (Dyngjusandur), H55 (Hagavatn), Land1 (Landeyjarsandur), Maeli2 (Mælifellssandur) and MIR45 (Mýrdalssandur). As described in Baldo et al. (2020), 15 g of sediments (previously sieved to <1 mm) was placed in a Büchner flask, this was shaken for 5 min at 70 Hz on a sieve shaker (Retsch AS200), then mixed with nitrogen carrier gas and injected into the CESAM chamber at 10 L min$^{-1}$ for 10 min. Once injected in CESAM, the aerosolized dust particles were allowed to mix for around 10 min inside the chamber. A four-blade stainless-steel fan located at the bottom of the chamber ensured that the particle concentration was homogeneous inside the chamber. All experiments were conducted at ambient temperature and relative humidity < 2 %. A particle-free N$_2$/O$_2$ mixture was continuously injected into the chamber to balance the proportion of air flow sampled by the different instruments. The chamber was manually cleaned between experiments with different samples to avoid any cross-contamination. The background mass concentrations of dust aerosol in the chamber varied from 0.1 to 3.0 µg m$^{-3}$.

#### 2.1.1 Instrumentation

The physical and chemical properties of the suspended dust were monitored over the duration of the experiments (approximatively 3 hours) using different instruments connected to the chamber (Fig. S1).

The particle number size distribution of dust aerosols (in the units of cm$^{-3}$) was measured using a scanning mobility particle sizer (SMPS) (TSI Inc., DMA model 3080, CPC model 3772; 2.0 and 0.2 L min$^{-1}$ sheath–aerosol flow rates; 3-min time resolution), and two optical particle counters (OPCs), GRIMM (Grimm Inc., model 1.109; 655 nm operating λ; 1.2 L min$^{-1}$



flow rate; 6 s time resolution) and SkyGRIMM (Grimm Inc., model 1.129; 655 nm operating λ; 1.2 L min$^{-1}$ flow rate; 6 s time resolution). The SMPS measured the particle number concentration in the electrical mobility diameter ($D_m$) range from 0.019 to 0.870 µm. The OPC instruments GRIMM and SkyGRIMM measured the particle number concentration in the optical-equivalent diameter ($D_{op}$) range from 0.25 to 32 µm. The SMPS and SkyGRIMM were located at the bottom of the chamber, while the GRIMM was installed at the top of the chamber (about 1 m above the SMPS and the SkyGRIMM) to verify that the concentration of dust aerosols in the chamber was spatially uniform. The percentage difference between the number concentrations in the size range measured by GRIMM and SkyGRIMM was < 10% for the samples D3 and MIR45, < 5% for the samples Land1 and Maeli2, and < 2% for the sample H55 suggesting that homogeneous conditions were obtained during the experiments.

The scattering coefficient $\beta_{sca}$ (in the units of Mm$^{-1}$; Mm$^{-1}$ = 10$^{-6}$ m$^{-1}$) of the suspended particles was measured by a three-wavelength integrating nephelometer (TSI Inc., model 3563; operating at 450, 550, and 700 nm; 2 L min$^{-1}$ flow rate, 1 s time resolution). The absorption coefficient $\beta_{abs}$ (in the units of Mm$^{-1}$) was retrieved by a seven-wavelength aethalometer (Magee Sci., model AE31; operating at 370, 470, 520, 590, 660, 880, and 950 nm; 2 L min$^{-1}$ flow rate, 2-min time resolution). As detailed in Baldo et al. (2020), dust aerosol samples were also collected for 1 to 3 hours, each with a flow rate of 7 L min$^{-1}$, using custom-made filter samplers. The collected filter samples were analyzed offline to determine the chemical and mineralogical compositions of the suspended particles (Baldo et al., 2020).

The sampling system consisted of stainless-steel tubes (∼ 14-58 cm length, 9.5 mm inner diameter) located inside the chamber extracting air. For some instruments, conductive silicone tubing (TSI Inc., ∼ 30-65 cm length, 5 mm inner diameter) was also used as an external connection between the chamber and the instrument. The sampling lines were designed to be as straight and direct as possible to minimize particle loss. The total length of the sampling line varied depending on the instrument, 100 cm for the nephelometer, 86 cm for the aethalometer, 79 cm for the SMPS, 58 cm for the SkyGRIMM and the aerosol filter samplers, and 14 cm for the GRIMM.

To allow the comparison of measurements from different instruments, all data in the following analysis were averaged over 12-min intervals from the start of the filter sampling. The data analysis was conducted on the measurements starting after 30 min from the injection peak.

## 2.2 Data analysis

### 2.2.1 Calculation of the aerosol geometrical size distribution from SMPS and GRIMM measurements

The measurements from SMPS and GRIMM were combined into a single geometrical size distribution. Appendix A describes in detail the SMPS and GRIMM corrections. To combine SMPS and GRIMM measurements, first, the SMPS particle number concentration, $dN(D_m)/dlogD_m$, and GRIMM particle number concentration, $dN(D_{op})/dlogD_{op}$, were both converted into geometrical size distributions, $dN(D_{g,SMPS})/dlogD_{g,SMPS}$ and $dN(D_{g,GRIMM})/dlogD_{g,SMPS}$, respectively (see Fig. S2). For the SMPS, the volume-equivalent diameters ($D_g$) were retrieved from the mobility diameters $D_m$ using a range



of dynamic shape factors χ in Eq. (A1). The interval of χ values (from 1.6 to 2.0 by 0.1 steps) was selected according to the
aspect ratios observed in fine Icelandic dust particles ≤ 2.5 μm in diameter (Butwin et al., 2020). Iterative calculations were
performed to calculate $D_g$ using Eqs. (A1-3).

For the GRIMM optical diameters $D_{op}$, we used the correction factors calculated by Formenti et al. (2021) by applying the
Mie theory and assuming that the particles are homogeneous spheres. The $D_{op}/D_g$ range measured by the GRIMM is from
0.25 μm to 32 μm assuming the complex refractive index of polystyrene latex sphere (PSL) reference particles (n = 1.59 and
k = 0.000). For the Icelandic dust samples, we selected a range of potential complex refractive indices with the index n
varying from 1.57 to 1.63 by 0.01 steps and the index k varying from 0.000 to 0.020 by 0.001 steps. The range of n and k
values was chosen according to the initial guess of the complex refractive index of the Icelandic dust samples examined in
this study, estimated on the basis of their mineralogical composition (Baldo et al., 2020).

*Merging of the SMPS and GRIMM size distributions:* The SMPS geometrical size distributions, calculated assuming χ from
1.6 to 2.0 by 0.1 steps, and the GRIMM geometrical size distributions, calculated assuming 147 potential complex refractive
indices (n from 1.57 to 1.63 by 0.01 steps, and k from 0.000 to 0.020 by 0.001 steps), were combined in 735 different
geometrical size distributions $dN(D_g)/dlogD_g$. Figure S2 shows the good agreement between SMPS and GRIMM in their size
overlapping region. Since the size resolution of the SMPS measurements is higher than that of the GRIMM measurements,
we assumed the SMPS size distribution as a better representative in the overlapping region.

The merged geometrical size distributions were interpolated at a constant $dlogD_g$ interval of 1/64 using a fitted cubic
smoothing spline and normalized so that the total number of particles before and after interpolation is conserved. The
smooth.spline function from core CRAN R software was used to perform the interpolation following Beddows et al. (2010).
The total number of particles was calculated as follows:

$$N_{tot}(D_g) = \int_{D_{g,min}}^{D_{g,max}} \frac{dN(D_g)}{d \log D_g} \cdot d \log D_g \qquad (1)$$

After interpolation, the merged geometrical size distributions were corrected for the particle loss along the instrument
sampling lines. The particle loss calculator (PLC) software (von der Weiden et al., 2009) was used to calculate the particle
loss percentage for each instrument. The details of the method used in the particle loss correction process are described in
Appendix B. The results of this correction include the size distributions of the particles suspended in CESAM and the size
distributions of the particles sampled by the aethalometer and nephelometer. Since the size distributions of the two
instruments were very similar, we decided to estimate a common size distribution, calculated as the mean of the size
distributions of the aethalometer and nephelometer. The resulting size distribution, referred to as $N_{sw-OPAs}(D_g)$, represents the
size distribution of the Shortwave Optical Properties Analyzers (SW-OPAs).



Based on the combinations of χ, n, and k size corrections, a total of 735 geometrical size distributions were calculated. These were used to correct the spectral coefficients $\beta_{abs}$ and $\beta_{sca}$ measured respectively by the aethalometer and nephelometer, and

to retrieve the complex refractive index of Icelandic dust. An example of the procedure used to process the size distribution data is shown in Fig. S3.

Furthermore, two sensitivity simulations were performed to account for the potential error on measured particle number concentrations – $\sqrt{N(D_g)}$ according to the Poisson statistics – in the retrieval of the complex refractive index. The observed particle number concentrations plus or minus 1 standard deviation (SD), $N(D_g)$, $\pm \sqrt{N(D_g)}$, were used as upper and lower

limits of the SMPS and GRIMM measurements. In Test 1, corrections and calculations were performed using the SMPS and GRIMM data plus 1 SD uncertainty. In Test 2, we used the SMPS and GRIMM data minus 1 SD uncertainty.

Further uncertainty arises from the use of the Mie theory, which assumes homogeneous spherical particles. The particle morphology was not measured during the experiments, but previous research found that Icelandic dust particles can be irregular and present internal pores (Butwin et al., 2020; Richards-Thomas et al., 2020). In particular, the results of Butwin et

al. (2020) indicate that the morphological characteristics of Icelandic dust particles equal or less than 20 µm in diameter tend to be similar to those of mineral dust from global source regions, while larger particles exhibited morphological properties of volcanic ash. In this study, since the particle size distribution effectively measured by the GRIMM corresponds to around the PM$_{20}$ fraction, we can assume that Icelandic dust has similar morphological properties to typical mineral dust. Despite mineral dust largely exhibits irregular shapes (Huang et al., 2021), our calculations are based on homogeneous equivalent

spheres. However, previous studies have shown that not considering the asphericity of dust aerosols when converting $D_{op}$ into $D_g$ can lead to an underestimation of the coarse sizes (Formenti et al., 2021; Huang et al., 2021) as aspherical particles have a larger single-particle scattering cross-section $C_{sca}$ compared to the volume-equivalent spheres. This effect is greater as the imaginary index k of the dust particles increases (Huang et al., 2021). The uncertainty resulting from using the Mie theory was not estimated in this study.


*Calculation of the effective diameter:* The effective diameter ($D_{eff}$) can be used as a measure of the aerosol size distribution as defined by Eq. (2) (Hansen, 1971):

$$D_{eff} = \frac{\int_{D_{g,min}}^{D_{g,max}} D_g^3 \cdot \frac{dN(D_g)}{d\log D_g} \cdot d\log D_g}{\int_{D_{g,min}}^{D_{g,max}} D_g^2 \cdot \frac{dN(D_g)}{d\log D_g} \cdot d\log D_g} \tag{2}$$

$D_{eff}$ differs from the simple mean diameter because the particle surface area is included as a weight factor. Since light scattering is proportional to the particle surface area, $D_{eff}$ is relevant for describing the scattering properties of size

distributions (Hansen, 1971). In this study, $D_{eff}$ was calculated for the aerosol fine fractions ($D_g \leq 1$ µm), $D_{eff,fine}$, and the coarse fractions ($D_g > 1$ µm), $D_{eff,coarse}$, by varying the input parameters χ, n, and k in the examined ranges.



### 2.2.2 Spectral scattering coefficient measured by the nephelometer

The aerosol scattering coefficient $\beta_{sca}$ was measured by the nephelometer at 450, 550, and 700 nm. The nephelometer can only detect scattering angles ($\theta$) from about 7º to 170º rather than from 0º to 180º causing a systematic underestimation of

the total scattering known as angular truncation error (Anderson et al., 1996; Anderson and Ogren, 1998). The truncation correction $C_{trunc}$ for total scattering was estimated from Mie calculations using $N_{sw-OPAs}(D_g)$ as input as described in Appendix C.

The average of the $C_{trunc}(\lambda)$ values obtained for different $N_{sw-OPAs}(D_g)$ was used to correct the data. The $C_{trunc}$ decreased with increasing $\lambda$, and with the dust residence time in the chamber following the decrease in the contribution of the coarse fraction

to the aerosol size distribution. $C_{trunc}(\lambda)$ was similar for different Icelandic dust samples and varied between 1.3-1.9. These results are comparable to the range of $C_{trunc}$ values (from 1.2 to 1.7) reported by Di Biagio et al. (2019). The uncertainty on $C_{trunc}(\lambda)$ was estimated from the relative standard deviation (RSD) of $C_{trunc}(\lambda)$ obtained for different $N_{sw-OPAs}(D_g)$. The uncertainty on $C_{trunc}(\lambda)$ was in the range 7-17%.

The scattering coefficient at a given $\lambda$ was corrected by multiplying $\beta_{sca}$ by $C_{trunc}$. The uncertainty on $\beta_{sca}$ at 450, 550, and

700 nm was calculated using the error propagation method considering the photon counting and gas calibration uncertainty (5%), the SD over 12-min intervals, and the uncertainty on $C_{trunc}$. The uncertainty on $\beta_{sca}$ corrected for truncation varied between 9% and 19%. Once corrected for truncation, $\beta_{sca}$ was extrapolated at the aethalometer operating $\lambda$ (370, 470, 520, 590, 660, 880, and 950 nm) using the scattering Ångström exponent ($\mathring{a}_{sca}$) which was obtained from the power-law fitting of $\beta_{sca}(\lambda)$ versus $\lambda$ weighted by the inverse of the variance of $\beta_{sca}(\lambda)$. The uncertainty on $\beta_{sca}$ at 370, 470, 520, 590, 660, 880, and

950 nm was estimated as the quadratic combination of the average error on $\beta_{sca}$ corrected for truncation and the root-mean-square error (RMSE) of $\beta_{sca}$ predicted at 450, 550, and 700 nm. The uncertainty on $\beta_{sca}$ predicted was 11%-18%.

### 2.2.3 Spectral absorption coefficient measured by the aethalometer

The aerosol absorption coefficient $\beta_{abs}$ was retrieved from the aethalometer measurements at 370, 470, 520, 590, 660, 880, and 950 nm. The aethalometer measures the spectral attenuation (ATTN) of light passing through an aerosol-laden quartz

filter, which is then converted into the attenuation coefficient (e.g., Weingartner et al., 2003):

$$ATTN = \ln\frac{I_0}{I} \qquad (3)$$

where $I_0$ is the intensity of light passing through the blank portion of the filter, and $I$ is the intensity of light passing through the loaded filter. From the measured ATTN the attenuation coefficient ($\beta_{ATTN}$, m$^{-1}$) is retrieved as:





$$\beta_{ATTN}(\lambda) = \frac{\Delta ATTN(\lambda)}{\Delta t} \cdot \frac{A}{V} \tag{4}$$

where A is the area of the aerosol collection spot ($0.00005 \pm 0.00001$ m$^2$), V is the sampling flow rate ($0.002$ m$^3$ min$^{-1}$), and $\Delta ATTN$ ($ATTN_n - ATTN_{n-1}$) is the variation of ATTN over 2-min time intervals ($\Delta t$) that corresponds to the time resolution of the aethalometer.

The spectral coefficient $\beta_{ATTN}$ contains the contributions of light absorption by the aerosol particles deposited on the filter and is affected by scattering by the filter material itself. The aerosol absorption coefficient $\beta_{abs}$ was obtained by correcting $\beta_{ATTN}$ for measurements artefacts according to the correction scheme from Collaud Coen et al. (2010):

$$\beta_{abs}(\lambda) = \frac{\beta_{ATTN}(\lambda) - \alpha(\lambda) \cdot \beta_{sca}(\lambda)}{R(\lambda) \cdot C_{ref}(\lambda)} \tag{5}$$

where $\alpha(\lambda)\beta_{sca}(\lambda)$ is the scattering effect correction. $\beta_{sca}(\lambda)$ is the scattering coefficient weighted by the $\alpha(\lambda)$ parameter. This term represents the scattered radiation by the aerosol particles deposited on the filter and miscounted as attenuation. $C_{ref}(\lambda)$ is the multiple scattering effect correction, representing multiple scattering by the filter fibers. $R(\lambda)$ is the loading effect correction, which accounts for the reduced aethalometer response with time due to light-absorbing particles accumulating on the filter. The details of the calculations of the parameters in Eq. (5) are described in Appendix D.

The uncertainty on $\beta_{abs}(\lambda)$ corrected according to Eq. (5) was calculated using the error propagation method considering the uncertainty on $\alpha(\lambda)$, $\beta_{sca}(\lambda)$, $C_{ref}$, and $R(\lambda)$. The uncertainty on $\beta_{abs}(\lambda)$ as 2-min intervals varied between 25%-78%. Ultimately, $\beta_{abs}(\lambda)$ was averaged over 12-min intervals, and the final uncertainty was calculated as the quadratic combination of SD and the average systematic error over 12-min intervals. The uncertainty on $\beta_{abs}(\lambda)$ as 12-min intervals varied between 27%-75%.

## 2.3 Derivation of the aerosol optical properties

### 2.3.1 Calculation of the aerosol spectral single scattering albedo

The spectral coefficients $\beta_{sca}(\lambda)$ and $\beta_{abs}(\lambda)$ obtained from the nephelometer and aethalometer measurements were used to calculate 12-min average values of the extinction coefficient $\beta_{ext}(\lambda)$ in Mm$^{-1}$ at 370, 470, 520, 590, 660, 880, and 950 nm:

$$\beta_{ext}(\lambda) = \beta_{sca}(\lambda) + \beta_{abs}(\lambda) \tag{6}$$

SSA($\lambda$) was also calculated at a 12-min resolution:



$$SSA(\lambda) = \frac{\beta_{sca}(\lambda)}{\beta_{sca}(\lambda) + \beta_{abs}(\lambda)} \quad (7)$$

The uncertainty on $\beta_{ext}(\lambda)$ was 10%-17%, while the uncertainty on $SSA(\lambda)$ varied between 15%-25%, which were estimated

by the error propagation through Eq. (6)(6) and Eq. (7)(6), respectively, considering the uncertainty on $\beta_{sca}(\lambda)$ and $\beta_{abs}(\lambda)$.

In addition, the experiment-averaged single scattering albedo $SSA_{avg}$ at 370, 470, 520, 590, 660, 880, and 950 nm was retrieved from the slope ($m_{RMA}$) of the linear regression between $\beta_{sca}(\lambda)$ and $\beta_{abs}(\lambda)$ starting from 30 min after the dust injection peak to 2.5 h (Di Biagio et al., 2019; Moosmuller et al., 2012):

$$SSA_{avg}(\lambda) = \left(1 + \frac{1}{m_{RMA}(\lambda)}\right)^{-1} \quad (8)$$

The linear fitting was performed using the reduced major axis (RMA) regression, because both variables (x and y) come

from measurements and are subject to errors (Ayers, 2001; Smith, 2009). Overall, a strong correlation between $\beta_{sca}(\lambda)$ and $\beta_{abs}(\lambda)$ was observed ($R^2 > 0.99$). The uncertainty on $SSA_{avg}(\lambda)$ calculated considering the error on $m_{RMA}$ in Eq. (8) was $\leq$ 8%.

### 2.3.2 Retrieval of the spectral complex refractive index

The spectral coefficients $\beta_{abs}(\lambda)$ and $\beta_{sca}(\lambda)$ were computed using the SW-OPA geometrical size distributions $N_{sw\text{-}OPAs}(D_g)$ for

the 735 different combinations of the input parameters (X, n, and k), and hereinafter referred to as $\beta_{abs,model}(\lambda)$ and $\beta_{sca,model}(\lambda)$, respectively (see Appendix C). Subsequently, the complex refractive index at 370, 470, 520, 590, 660, 880, and 950 nm was determined through the comparison between $\beta_{abs,model}(\lambda)$ and $\beta_{sca,model}(\lambda)$ and those measured by the nephelometer and aethalometer referred to as $\beta_{abs,meas}(\lambda)$ and $\beta_{sca,meas}(\lambda)$, at individual time points throughout the time series. For each $\beta_{abs,model}(\lambda)$ and $\beta_{sca,model}(\lambda)$ scenario, we estimated the model error as the percentage difference (%diff) respectively

with $\beta_{abs,meas}(\lambda)$ and $\beta_{sca,meas}(\lambda)$. To account for the uncertainty on $\beta_{abs,meas}(\lambda)$ and $\beta_{sca,meas}(\lambda)$, we used $\beta_{abs,meas}(\lambda) \pm 1$ SD and $\beta_{sca,meas}(\lambda) \pm 1$ SD as the upper and lower limits of the measurements. We selected $\beta_{abs,model}(\lambda)$ and $\beta_{sca,model}(\lambda)$ scenarios with the lowest model error corresponding to 0.1-0.5 quantile of %diff. Subsequently, the results for $\beta_{abs,meas}(\lambda) \pm 1$ SD and $\beta_{sca,meas}(\lambda) \pm 1$ SD (consisting of three datasets for each spectral optical coefficient) were examined, and only the $\beta_{abs,model}(\lambda)$ and $\beta_{sca,model}(\lambda)$ scenarios common to all six datasets were kept. The indices $k(\lambda)$ and $n(\lambda)$ were retrieved from the selected

scenarios. Since the comparison between calculation and measurements resulted in multiple solutions for $k(\lambda)$ and $n(\lambda)$, we calculated the mean of the k solutions and of the n solutions.

In addition, the experiment-averaged spectral complex refractive index was determined based on the linear fit between $\beta_{abs,meas}(\lambda)$ vs $\beta_{abs,model}(\lambda)$, and $\beta_{sca,meas}(\lambda)$ vs $\beta_{sca,model}(\lambda)$ scenarios starting from 30 min after the dust injection peak to 2.5 h. To retrieve the experiment-averaged real index $n_{avg}(\lambda)$ and imaginary index $k_{avg}(\lambda)$, we updated the method applied to





determine $k(\lambda)$ and $n(\lambda)$ at a 12-min resolution. For each $\beta_{abs,model}(\lambda)$ and $\beta_{sca,model}(\lambda)$ scenarios, we assumed that the input parameters $\chi$, $k(\lambda)$ and $n(\lambda)$ are constant throughout the experimental run. From the linear fit between $\beta_{abs,meas}(\lambda) \pm 1$ SD and $\beta_{abs,model}(\lambda)$ scenarios, and between $\beta_{sca,meas}(\lambda) \pm 1$ SD and $\beta_{sca,model}(\lambda)$ scenarios, we selected only the model estimates which showed a high correlation with observations ($R^2 > 0.70$). The modelled and measured spectral coefficients were then compared based on the root mean square error (RMSE) instead of using the %diff at individual time points. The uncertainty on $k_{avg}(\lambda)$ and $n_{avg}(\lambda)$ was estimated from the RSD of the k and n solutions. The uncertainty on $k_{avg}(\lambda)$ was up to 99%, while the uncertainty on $n_{avg}(\lambda)$ was < 2%.

These calculations were repeated in Test 1 and Test 2, the sensitivity simulations to account for the uncertainty resulting from the size distribution measurements which were used to correct the spectral coefficients measured by the aethalometer and nephelometer, and to calculate the $\beta_{abs,model}(\lambda)$ and $\beta_{sca,model}(\lambda)$ scenarios.

## 3 Results

### 3.1 Aerosol size distribution and effective diameter

Figure S2 shows examples of geometrical size distributions that were obtained using various dynamic shape factors $\chi$ and complex refractive indices for the dust particles. For spherical particles ($\chi = 1$ and $D_m = D_g$), the $D_g$ range measured by the SMPS was from 0.019 µm to 0.87 µm. For non-spherical particles ($\chi > 1$), $D_g$ decreased with increasing $\chi$ as defined by Eq. (A1). After the conversion of $D_m$ to $D_g$, the $D_g$ range of the SMPS was around 0.015-0.58 µm for $\chi = 1.6$, 0.015-0.56 µm for $\chi = 1.7$, 0.014-0.53 µm for $\chi = 1.8$, 0.014-0.51 µm for $\chi = 1.9$, and 0.013-0.49 µm for $\chi = 2.0$.

The correction factors used to convert $D_{op}$ into $D_g$ had a considerable impact on $D_g > 0.6$ µm. $D_g$ increased with k, while the variability of n mainly affected $D_g$ at around 1 µm. The minimum $D_g$ varied between around 0.24-0.25 µm. For $k \leq 0.003$, the maximum $D_g$ sharply increased with k from 31.5-32.5 µm to 85.6-90.4 µm. For $k > 0.003$, the maximum $D_g$ showed only a small variation reaching up to around 94 µm (Fig. S4).

The overlapping interval between the SMPS and GRIMM data decreased as $\chi$ increased. The GRIMM size distributions tend to spread out between 0.6-2 µm (Fig. S2), consequently reducing the smoothness of the fitted size distributions within this size range, which is likely due to the larger uncertainty of the correction factors at these $D_g$ values (Formenti et al., 2021). After correcting the data for the loss along the instrument sampling lines, the size distribution of the particles suspended in CESAM was in the $D_g$ range up to 20 µm, while the size distribution of the particles sampled by the SW-OPAs was in the $D_g$ range up to 9 µm.

The average of the $D_{eff}$ values derived from different $\chi$-n-k combinations was reported as the $D_{eff}$ of the Icelandic dust samples, with the RSD as the uncertainty on $D_{eff}$. The effective diameter of the coarse fractions, $D_{eff,coarse}$, showed a strong positive correlation with the input parameter k ($R^2 = 0.6$-0.8), while the correlation between $D_{eff,coarse}$ and $\chi$ or n was low ($R^2 < 0.1$). The correlation between the effective diameter of the fine fractions, $D_{eff,fine}$, and all input parameters was low. Figure S5 shows an example of the correlation between $D_{eff}$ determined from the SW-OPA size distributions and $\chi$, n, and k.





The parameter $D_{eff}$ was used as an indication of the stability of the size distribution in the chamber. Figure 2 shows an example of how $D_{eff}$ changed over time. The $D_{eff,coarse}$ decreased with time due to the rapid deposition of the largest particles in the chamber. In Icelandic dust samples, $D_{eff,coarse}$ calculated using the SW-OPA size distributions varied from ~3.1-3.5 µm

after around 30 min from the injection peak to 2.5-2.7 µm after around 2.5 h from the injection peak. For the particles suspended in CESAM, $D_{eff,coarse}$ varied from 8.4-11 µm (30 min after the injection peak) to 3.7-4.4 µm (2.5 h after the injection peak). The $D_{eff,fine}$ remained relatively constant over the duration of the experiments varying between 0.5-0.7 µm in different samples. The uncertainty on $D_{eff,coarse}$ was < 14%-19% for the SW-OPA size distributions, and 44%-56% for the particles suspended in CESAM, while the uncertainty on $D_{eff,fine}$ was less than 3%. The $D_{eff}$ values for all the samples are

shown in Figs. 2, and Figs. S6-10.

The $D_{eff}$ results from the sensitivity studies (Tests 1-2) to account for the error on the SMPS and GRIMM measurements were consistent within the uncertainties with the results from the base simulation (Fig. 2 and Figs. S6-10). The difference between the $D_{eff}$ results from the sensitivity studies and $D_{eff}$ from the base simulation was not significant because it was less than three times the square root of the sum of their squared uncertainties. For the SW-OPA size distributions, $D_{eff,coarse}$

decreased from ~3.3-3.6 µm (30 min after the injection peak) to 2.8-3.0 µm (2.5 h min after the injection peak) in Test 1, and from 2.8-3.3 (30 min after the injection peak) to 1.4-1.9 (2.5 h min after the injection peak) in Test 2. For the particles suspended in CESAM, $D_{eff,coarse}$ decreased from ~10-12 µm (30 min after the injection peak) to 4.6-5.7 µm (2.5 h min after the injection peak) in Test 1, and from 3.3-5.8 (30 min after the injection peak) to 1.6-2.0 (2.5 h min after the injection peak) in Test 2. The uncertainty on $D_{eff,coarse}$ was 12%-17% in Test 1 and 9%-22% in Test 2 for the SW-OPA size distributions, and

41%-55% in Test 1 and 9%-50% in Test 2 for the particles suspended in CESAM. The results of $D_{eff,fine}$ were not considerably affected by the sensitivity studies.

### 3.2 Spectral extinction and absorption coefficients, single scattering albedo, and complex refractive index

Figure 3 presents the temporal variation of $\beta_{ext}(\lambda)$, $\beta_{abs}(\lambda)$, and SSA($\lambda$) of the Icelandic dust samples, as derived from the base simulation. The coefficients $\beta_{ext}(\lambda)$ and $\beta_{abs}(\lambda)$ decreased with $\lambda$, with the largest variation observed for $\beta_{abs}(\lambda)$ between 370

and 590 nm. On the other hand, SSA($\lambda$) increased marginally with $\lambda$, within the error bars. Both $\beta_{ext}(\lambda)$ and $\beta_{abs}(\lambda)$ decreased with time at all values of $\lambda$, while SSA($\lambda$) was relatively constant. The results of the sensitivity studies for $\beta_{ext}(\lambda)$, $\beta_{abs}(\lambda)$, and SSA($\lambda$) were in agreement with the results of the base simulation within their respective uncertainties (Figs. S11-15).

Figure 4 provides an example of the temporal variation of $n(\lambda)$ and $k(\lambda)$ in two Icelandic dust samples, including the results from both the base simulation and the sensitivity studies. The parameter $n(\lambda)$ showed no dependence on $\lambda$ and time and

varied within the range of examined n values (from 1.57 to 1.63 by 0.01 steps) with an uncertainty of < 3% in the base simulation, Test 1, and Test 2 (Figs. S16-20). For the parameter $k(\lambda)$, the data at 12-min resolution were too noisy to identify a clear relationship with time or $\lambda$ (Figs. S16-20). In Test 1, $k(\lambda)$ was almost constant with time and $\lambda$. In Test 2, k did not show a clear dependence on $\lambda$. For the sample MIR45, $k(\lambda)$ exhibited an increasing trend over time. For D3 and Maeli2, $k(\lambda)$ increased after around 1.5 h from the dust injection peak. For H55 and Land1, $k(\lambda)$ did not show a temporal trend. The



uncertainty on k(λ) at 12-min resolution varied up to 99% in the base simulation, 198% in Test 1, and 79% in Test 2. We compared k(λ) and $D_{eff,coarse}$ to determine the impact of the reduction in the particle coarse fraction over time on the complex refractive index. However, a clear correlation between $D_{eff,coarse}$ and k(λ) was not observed. The k(λ) results at a 12-min resolution were noisy, making it challenging to establish a meaningful relationship between the two parameters, as the slopes of the regression lines between $D_{eff,coarse}$ and k(λ) were close to zero.

The experiment-averaged single scattering albedo $SSA_{avg}$ increased with λ from 370 to 590 nm, though with some uncertainty, but remained relatively constant between 590 and 950 nm (Fig. 5). The $SSA_{avg}(λ)$ varied from 0.93 at 370 nm to 0.96 at 950 nm for the sample D3, from 0.94 to 0.96 for H55, from 0.91 to 0.96 for Land 1, from 0.90 to 0.95 for Maeli2, and from 0.90 to 0.94 for MIR45 (Table 1). The uncertainty on $SSA_{avg}(λ)$ was ≤ 8%. The $SSA_{avg}(λ)$ results from the sensitivity studies were within the uncertainty for the base simulation (Table S1).

The experiment-averaged imaginary index, $k_{avg}(λ)$, decreased with λ, from 0.006 at 370 nm to 0.002 at 950 nm for the sample D3, from 0.005 to 0.003 for H55, Land1, and Maeli2, and from 0.005 to 0.003 for MIR45, as determined from the base simulation (Table S2). $k_{avg}(λ)$ was sensitive to the particle size distribution. The variation of $k_{avg}$ with respect to λ was less evident in Test 1 compared to the base simulation (Table S2). The $k_{avg}(λ)$ values varied between 0.003 and 0.001 for H55 and Land1, while they remained relatively constant at around 0.002 for Maeli2 and MIR45. For the sample D3, $k_{avg}(λ)$

was 0.002 at 370 nm and around 0.001 for all the other λ. In Test 2, $k_{avg}$ increased with λ from 0.002 at 370 nm to 0.007 at 950 nm for the sample H55. For all the other Icelandic dust samples, the relationship between $k_{avg}$ and λ was not monotonic, and $k_{avg}$ varied in the range between 0.002-0.007 for D3, 0.003-0.006 for Land1, 0.002-0.005 for Maeli2, and 0.005-0.007 for MIR45. Overall, $k_{avg}(λ)$ estimates of Test 1 were lower than those of the base simulation and Test 2 (Table S2). The uncertainty on $k_{avg}(λ)$ was generally < 85% in the base simulation and Test 1, and < 70% in Test 2.

The experiment-averaged real index, $n_{avg}(λ)$, showed no trend with λ, and $n_{avg}(λ)$ varied in the range between 1.59-1.62 in the base simulation, 1.58-1.61 in Test 1, and 1.57-1.63 in Test 2 (Table S3). Overall, the uncertainty on $n_{avg}(λ)$ was < 2%.

Table S4 reports a summary of the comparison between $SSA_{avg}(λ)$ obtained from the measured and computed spectral coefficients for the base simulation, Test 1, and Test 2. Although the RMSE values were generally low, the correlation between the measured and modelled $SSA_{avg}(λ)$ tends to be higher in the base simulation and Test 1 compared to Test 2

(Table S4). Di Biagio et al. (2019) chose to average the k values from all three scenarios. Here the increase of $k_{avg}$ with λ in Test 2 is hard to explain, suggesting that Test 2 results are not realistic. Based on this, we chose to combine the results from the base simulation and Test 1 to obtain a single set of values for k(λ) (Table 2) and n(λ) (Table 3).

## 4 Discussion

The Icelandic dust samples examined in this study showed similar spectral optical properties. The spectral single scattering

albedo SSA(λ) (Table 1) and imaginary index k(λ) (Table 2) showed opposite trends, as expected. SSA(λ) increased from



0.90-0.94 at 370 nm to 0.94-0.96 at 950 nm in different samples, while k(λ) decreased slightly from 0.004 at 370 nm to 0.002-0.003 at 950 nm. The real index n(λ) was rather constant with λ at 1.60-1.61 (Table 3).

**4.1 Spectral complex refractive index and single scattering albedo of Icelandic dust comparison with literature data, volcanic ash, and low-latitude dust**

The complex refractive index of Icelandic dust is consistent with the initial estimate of Baldo et al. (2020) based on the volume-averaged mineralogical composition and reference complex refractive indices of individual minerals. In Baldo et al. (2020), n(λ) ranged from 1.57-1.63 between 470 and 660 nm, while k(λ) was sensitive to the reference refractive indices of magnetite applied in the calculations, and varied up to one order of magnitude in different absorption scenarios. In this study, n(λ) (Table 3) is comparable to the average value of the initial estimate, and k(λ) (Table 2 and Fig. 6) is within or at the
lower end of the range of k(λ) values estimated by Baldo et al. (2020).

The value of k at 660 nm of the Icelandic dust sample from the Mýrdalssandur hotspot (southern Iceland) is lower than that reported in the literature by Zubko et al. (2019). In Zubko et al. (2019), the complex refractive index of Icelandic dust was retrieved based on the measurements of the angular scattered-light intensity and degree of linear polarization of aerosol particles generated from volcanic sand from the Mýrdalssandur area. They estimated a value of 1.60 for n and 0.01 for k at
647 nm. (Zubko et al., 2019), the latter is around 3 times higher than the value observed in this study at 660 nm (Table 2).

Icelandic dust shares a similar origin with volcanic ash; however, it is difficult to assess the complex refractive index of Icelandic dust based on what has been observed for Icelandic volcanic ash. This is because the literature reports a wide range of values for k and n in the SW-spectrum for Icelandic ash (Fig. 7). The values of n(λ) for Icelandic ash vary from 1.50 to 1.65 (Bukowiecki et al., 2011; Deguine et al., 2020; Derimian et al., 2012; Reed et al., 2018; Weinzierl et al., 2012).
Laboratory studies conducted on re-suspended ash samples found that k(λ) of Icelandic ash remains below 0.001 in the UV-visible spectrum and tend to increase towards the near-IR region (Deguine et al., 2020; Reed et al., 2018). The k(λ) values resulting from direct measurements of the Eyjafjallajökull plume in 2010 (e.g., Bukowiecki et al., 2011; Derimian et al., 2012; Weinzierl et al., 2012) are considerably different from the results of laboratory measurements (Deguine et al., 2020; Reed et al., 2018). In Derimian et al. (2012), the range of k(λ) values for Eyjafjallajökull ash varied from around 0.015 at
440 nm to 0.008 at 870 nm, which is similar to what reported by Weinzierl et al. (2012), whereas Bukowiecki et al. (2011) reported an average value of 0.02 between 450-700 nm. Here, n(λ) of Icelandic dust (Table 3) falls within the range of n values reported for Icelandic ash (Fig. 7a) (Bukowiecki et al., 2011; Deguine et al., 2020; Derimian et al., 2012; Reed et al., 2018; Weinzierl et al., 2012), while the values of k(λ) of Icelandic dust (Table 2) are higher than the k(λ) values of Icelandic ash from laboratory measurements (Deguine et al., 2020; Reed et al., 2018), but lower than ambient observations
(Bukowiecki et al., 2011; Derimian et al., 2012; Weinzierl et al., 2012) (Fig. 7b). The difference between observational data and laboratory measurements could be due to several factors, including, for example, the different methods used, and the different type of samples analyzed, as the physical and chemical properties of re-suspended volcanic ash deposited in situ could be different from those of volcanic ash plumes, which could change further during atmospheric transport. On the other





hand, our results for the SSA(λ) of Icelandic dust (Table 1 and Fig. 5) are within the range of values observed in Icelandic ash plumes (e.g., Derimian et al., 2012; Toledano et al., 2012). During the 2010 eruptions of Eyjafjallajökull, the values of SSA(λ) of a volcanic ash plume over France (17 April 2010) varied from 0.81 at 440 nm to 0.92 at 870 nm from Aerosol Robotic Network (AERONET) observations (Derimian et al., 2012), whereas SSA(λ) derived from AERONET measurements of the ash plumes over Spain and Portugal (6-12 May 2010) was found to be 0.95 at 440 nm but slightly decreased with λ (Toledano et al., 2012).

In Fig. 8, the complex refractive index and SSA(λ) of Icelandic dust were compared with those of low-latitude dust from major source regions such as northern Africa-Sahara, Sahel, and Eastern Asia (Di Biagio et al., 2019). The real index n(λ) of Icelandic dust, at around 1.6 (Table 3), is slightly higher than that of typical low-latitude dust, averaging 1.51 in the spectral range 370-950 nm (Di Biagio et al., 2019) (Fig. 8a). Fig. 8b shows that the mean value of k(λ) of Icelandic dust (Table 2) is at the upper end of the range of typical low-latitude dust (Di Biagio et al., 2019). The values of k(λ) decreased from around

0.002-0.005 at 370 nm to 0.001 at 950 nm in northern African-Saharan dust, from 0.001-0.009 at 370 nm to 0.0003-0.002 at 950 nm in Sahelian dust, and from 0.002-0.004 at 370 nm to 0.0005-0.001 at 950 nm in eastern Asian dust (Di Biagio et al., 2019). In the spectral range between 370 and 590 nm, the difference of k(λ) of Icelandic dust (Table 2) with the k(λ) values of northern African-Saharan dust, Sahelian dust, and eastern Asian dust was less than three times the square root of the sum of their squared uncertainties. The largest difference between Icelandic dust and low-latitude dust samples was observed

from 660 to 950 nm (Fig. 8b). In particular, at these λ, the values of k(λ) of the samples D3 and MIR45, representative of the two major dust hotspots in Iceland (e.g., Arnalds et al., 2016), showed a significant difference compared to several low-latitude dust, and were around 2-5 times higher than the k(λ) values of mineral dust from Morocco, Libya, and Algeria in northern Africa-Sahara, and the Taklimakan Desert in eastern Asia, and 5-8 times higher than the k(λ) values of dust from the Bodele Depression in the Sahel (Fig. 8b).

Although SSA(λ) of Icelandic dust (Table 1 and Fig. 5) is within the range of values of low-latitude dust (Di Biagio et al., 2019), an absorption of 4%-6% can still be observed at 950 nm (Fig. 8c), which is relatively high, despite the uncertainty, considering that absorption in mineral dust at these λ is generally around 2% (Di Biagio et al., 2019). The SSA is affected by particle size and chemical composition. The particle size of Icelandic dust is quite comparable to that of low-latitude dust (Di Biagio et al., 2019), according to the $D_{eff}$ values for both the coarse and fine fractions of the particles sampled by the

SW-OPAs. This suggests that the differences between the two may be mainly related to mineralogy.

## 4.2 Optical properties and mineralogy

The chemical and mineralogical composition of the Icelandic dust samples examined in this study was determined by Baldo et al. (2020). Icelandic dust showed a basaltic composition, with relatively high total Fe content varying from around 11 wt% in the sample H55 to 13 wt% in MIR45. The samples were primarily composed of amorphous basaltic materials ranging

from 8 wt % in H55 to 60 wt %-90 wt % in the other Icelandic dust. MIR45 had the highest proportion of amorphous glass. The magnetite fraction (1 wt%-2 wt %) was considerably higher than in low-latitude dust contributing to 7 %-15 % of total



Fe. Fe oxide minerals accounted for 9%-20% of the total Fe, while most of the iron was contained in other mineral phases (e.g., pyroxene, feldspars and olivine) and in the amorphous glass (Baldo et al., 2020).

The differences between Icelandic dust and low-latitude dust could be related to the presence of different Fe oxide minerals.

Hematite ($Fe_2O_3$) and goethite (FeO(OH)) are the main Fe oxide in low-latitude dust (Di Biagio et al., 2019; Formenti et al., 2014) and have specific absorption bands in the UV-visible spectrum (Caponi et al., 2017; Derimian et al., 2008; Di Biagio et al., 2019; Dubovik et al., 2002; Engelbrecht et al., 2016; Formenti et al., 2014; Lafon et al., 2006; Moosmuller et al., 2012; Redmond et al., 2010). However, we have shown that Icelandic dust is rich in magnetite (Baldo et al., 2020) which also absorbs light in the UV-visible spectrum, but has relatively high absorption even in the near-IR with k > 0.2 at 950 nm

(Table S5). Di Biagio et al. (2019) found a linear relationship between SSA($\lambda$) and k($\lambda$) and the mass concentrations of Fe oxides and total Fe. In this study, the sample size was too small to reach statistical significance, and the values of SSA($\lambda$) and k($\lambda$) were similar among different samples (Tables 1-2) and did not show a clear pattern. In addition, since amorphous basaltic material is dominant, but has main absorption bands in the IR region (Deguine et al., 2023; Deguine et al., 2020), we do not expect a good correlation. Overall, the values of k($\lambda$) of Icelandic dust samples (Table 2) appear to be more similar to

those of mineral dust from the Sahel region in Niger and Mali (Fig. 8b), which have relatively high Fe oxide content, 5.8 wt% and 3.7 wt%, respectively (Di Biagio et al., 2019). However, it is important to note that the total Fe oxide content in Icelandic dust is lower, ranging from 1.2 wt%-2.6 wt% (Baldo et al., 2020). Additionally, it is worth noting that the Bodele sample was classified as part of the Sahel in Di Biagio et al. (2019), but samples from the Bodele Depression are often dominated by diatoms, with extremely low Fe oxide content.(e.g., Bristow et al., 2009; Shi et al., 2011).

The high magnetite content may contribute to the high absorption capacity of Icelandic dust. In Fig. 6, the k($\lambda$) values estimated in this study tend to be closer to the volume-averaged imaginary indices calculated by Baldo et al. (2020) assuming the spectral refractive index of magnetite from Querry (1985) than to the values predicted using the refractive index of magnetite from Huffman and Stapp (1973). This suggest that the dataset from Querry (1985) may be more representative of the complex refractive index of magnetite in Icelandic dust. Table S5 reports the reference complex

refractive indices of mineral components of Icelandic dust in the spectral range 370-950 nm. The average value of k($\lambda$) of magnetite between 660 and 950 nm, 0.18 estimated using data from Querry (1985) and 0.43 using data from Huffman and Stapp (1973), is greater than the reference index of any other mineral components, up to 270 times and 640 times, respectively. This suggests that magnetite may be one of the main minerals contributing to light absorption in Icelandic dust between 660 and 950 nm.

**5 Concluding remarks: Implications for the radiative effect**

Here, we provide for the first time a dataset with the spectral SW- SSA (Table 1) and complex refractive index (Tables 2-3) of mineral dust from five major dust hotspots in Iceland. The spectral imaginary index k($\lambda$) of Icelandic dust (Table 2) is at



the upper end of the range of typical low-latitude dust. However, in Icelandic dust, absorption between 660 and 950 nm can be 2-8 times higher than in several low-latitude dust (Fig. 8b).

Previous research by Kylling et al. (2018) estimated that Icelandic dust produces a positive direct radiative forcing in the Arctic via dust-radiation interaction and deposition onto snow and ice. This estimation was based on the dust load estimates by Groot Zwaaftink et al. (2016) and assumed the same optical properties, in line with the dataset of Balkanski et al. (2007), for all mineral dust from different regions. The dust radiative effect is very sensitive to the complex refractive index, which is a major uncertainty (e.g., Di Biagio et al., 2020; Li et al., 2021). In the dataset used by Kylling et al. (2018), $k(\lambda)$ was

around 0.002 at 555 nm, which is comparable to the values of $k(\lambda)$ of Icelandic dust at 590 nm observed in this study (Table 2). However, $k(\lambda)$ in the Icelandic dust samples was found to be 0.002-0.003 in the range of 660 to 950 nm (Table 2), while the results from Balkanski et al. (2007) indicated that $k(\lambda)$ decreases to below 0.001 in the same $\lambda$ range. This implies that the positive direct radiative forcing of Icelandic dust in the Arctic could be stronger than previously estimated. The measurements conducted at CESAM on the longwave optical properties of Icelandic dust will provide additional valuable

insights into its radiative effect and contribution to Arctic warming. As glaciers retreat and expose more ice-free land as potential dust sources (Meinander et al., 2022), this effect is expected to increase. While these new results on the optical properties of Icelandic dust will contribute to reducing the uncertainty on its radiative effect, additional data on the size distribution and emissions are also crucial to advance our understanding.




**Appendix A: Retrieval of the aerosol size distribution**

The aerosol size distribution was obtained by combining the measurements from SMPS and GRIMM into a single geometrical size distribution. Several corrections were applied to the SMPS and GRIMM measurements because of the differences in operating principles and diameter definitions between the instruments.

The SMPS measures the size distribution of submicron particles. The operating principle of the SMPS is the balance between the electrical force on a charged particle due to a constant electric field and the drag force experienced by that particle at terminal velocity which is used to derive the particle size (DeCarlo et al., 2004; Hinds, 1999). The relationship between the electrical mobility diameter $D_m$ (the diameter of a single-charged spherical particle with the same terminal velocity in a constant electric field as the single-charged particle under consideration), as measured by the SMPS and the

volume-equivalent diameter $D_g$ (the diameter of a spherical particle of the same volume as the particle under consideration) is described by Eq. (A1):

$$D_g = \frac{D_m}{\chi} \cdot \frac{C_c(D_g)}{C_c(D_m)} \tag{A1}$$

where $\chi$ is the dynamic shape factor, $C_c$ is the Cunningham slip correction factor assuming a mean free path of 66 nm (air conditions) (Hinds, 1999):

$$D_{g\text{-}m} > 100 \text{ nm}, \qquad C_c(D_{g\text{-}m}) = 1 + \frac{66}{D_{g\text{-}m}} \cdot 2.52 \tag{A2}$$

$$D_{g\text{-}m} \leq 100 \text{ nm}, \qquad C_c(D_{g\text{-}m}) = 1 + \frac{66}{D_{g\text{-}m}} \cdot (2.34 + 1.05 \cdot e^{-0.39\frac{D_{g\text{-}m}}{66}}) \tag{A3}$$

where $D_{g\text{-}m}$ represents either the volume-equivalent diameter $D_g$ or the electrical mobility diameter $D_m$.

The OPCs measure the size distribution across a wide range of sizes including submicron and supermicron particles. The OPC operating principle is the dependence of angular light scattering on particle size (Hinds, 1999; Wendisch and Brenguier, 2013). The particle size measured by OPCs is therefore referred to as optical-equivalent diameter $D_{op}$, that is the diameter of a spherical particle with a given complex refractive index which has the same scattering efficiency ($Q_{sca}$) and cross-sectional area (A) as the particle under consideration. In this study, the volume-equivalent diameter $D_g$ of natural

aerosols was retrieved using look up tables from Formenti et al. (2021) which convert $D_{op}$ into $D_g$. These look up tables are calculated for a wide range of complex refractive indices to determine the diameter that would give the same (or closest) single-particle scattering cross-section, $C_{sca} = Q_{sca} \times A$, as that calculated using calibration reference PSL particles referring to $D_{op}$. For the GRIMM, increasing the imaginary index k, the scattering cross-section curve tends to become flat at around 1 µm as shown in Figure 2 of Formenti et al. (2021). This leads to large uncertainty when calculating the volume-equivalent



diameters $D_g$ of light-absorbing aerosols in the range between 0.6 and 2 µm (Formenti et al., 2021). Consequently, increasing the value of $D_{op}$ does not always correspond to increasing $D_g$, which may produce negative or very small $dlogD_g = logD_{g2} - logD_{g1}$, where $D_{g1}$ and $D_{g2}$ are the lower and upper cut-off diameters of each size range. This generates unusual spikes in the calculated geometrical size distributions $dN(D_g)/dlogD_g$. Two common strategies to address this issue are widening or grouping the OPC bin size (Johnson and Osborne, 2011) or fitting the corrected $D_g$ curve (Hand and Kreidenweis, 2002;

Johnson et al., 2008). Here, the corrected $D_g$ values which return $dlogD_g < 0.01$ were replaced with the average of the bin before and after.



## Appendix B: Particle loss correction

The particle loss correction accounts for the loss of particles along the instrument sampling lines. The particle loss

calculation was performed using the PLC software (von der Weiden et al., 2009). The input parameters for the PLC software included the geometry of the sampling line, the sampling flow rate, the particle density, and the dynamic shape factor $\chi$. The particle bulk density was set to $2.90 \pm 0.01$ g cm$^{-3}$ as measured using a helium pycnometer for particles less than 63 µm in diameter (Baldo et al., 2020), a value that is consistent with the results of a recent study by Richards-Thomas et al. (2020). However, from the analysis conducted on larger size fractions, less than 125 µm in diameter, using the water penetration

method, Butwin et al. (2020) found an average density value of 2.25 g cm$^{-3}$ for Icelandic dust. For each instrument, the particle loss percentage was calculated assuming $\chi$ from 1.6 to 2.0 by 0.1 steps.

To retrieve the size distribution of the particles suspended in CESAM, the merged geometrical size distributions were divided into two size ranges which were corrected for particle loss in the sampling systems of SMPS and GRIMM, respectively. The particle size domain was assigned based on the geometrical size distributions of GRIMM and SMPS used

to merge the data:

$$\frac{dN(D_g)}{d\log D_g} = \frac{dN(D_{g,SMPS})}{d\log D_{g,SMPS}} + \frac{dN(D_{g,GRIMM})}{d\log D_{g,GRIMM}} \tag{B1}$$

For the SMPS, the loss of particles with $D_g < 1$ µm was less than 5%. For the GRIMM, the particle loss was up to 15% in the range between 1-2.5 µm, within 50% in the range between 2.5-5 µm and reached 100% at $D_g$ around 20 µm. The variation in particle loss calculated using the examined $\chi$ values was generally below 10%, but over 50% at $D_g$ around 20 µm. The size distributions were corrected as follows:

$$\left[\frac{dN(D_g)}{d\log D_g}\right]_{Corr,SMPS-GRIMM} = \frac{dN(D_{g,SMPS-GRIMM})}{d\log D_{g,SMPS-GRIMM}} \cdot \frac{1}{1 - L(D_{g,SMPS-GRIMM})} \tag{B2}$$

where $dN(D_{g,SMPS-GRIMM})/d\log D_{g,SMPS-GRIMM}$ is the size distribution measured by the SMPS or GRIMM, and $L(D_{g,SMPS-GRIMM})$ is the proportion of particle lost as a function of $D_g$ in the size range measured by the SMPS or GRIMM. Finally, the size distribution of the particles suspended in CESAM was obtained summing up the contributions of the SMPS and GRIMM in their specific size range:

$$\left[\frac{dN(D_g)}{d\log D_g}\right]_{CESAM} = \begin{cases} \left[\dfrac{dN(D_g)}{d\log D_g}\right]_{Corr,SMPS} & ; \; SMPS_{min} < D_g < SMPS_{max} \\ \left[\dfrac{dN(D_g)}{d\log D_g}\right]_{Corr,GRIMM} & ; \; GRIMM_{min} < D_g < GRIMM_{max} \end{cases} \tag{B3}$$

Subsequently, the size distribution of the particles suspended in CESAM was used to evaluate the size-dependent correction

factor in the sampling lines of the aethalometer and nephelometer. For the aethalometer, the particle loss was below 5% at $D_g$



< 1µm, up to 20% in the range between 1-2.5 µm, below 60% in the range between 2.5-5 µm, and reached 100% at $D_g$ around 9 µm. The variation in particle loss calculated using the examined $\chi$ values was generally below 10%, but over 50% at $D_g$ around 9 µm. The percentage difference between the loss correction estimated for the aethalometer and that obtained for the nephelometer was < 11%. The data were corrected as follows:

$$\left[\frac{dN(D_g)}{d\log D_g}\right]_{Aet-Nep} = \left[\frac{dN(D_g)}{d\log D_g}\right]_{CESAM} \cdot \left(1 - L(D_g)_{Aet-Nep}\right) \tag{B4}$$

where $[dN(D_g)/d\log D_g]_{Aet-Nep}$ is the size distribution measured by the aethalometer or nephelometer and $L(D_g)_{Aet-Nep}$ is the particle loss as function of $D_g$ estimated for the two instruments. Since the size distributions of the aethalometer and nephelometer were very similar, we decided to estimate a common size distribution that would be representative of the dust particles sampled by these instruments. $N_{sw-OPAs}(D_g)$ is the size distribution of the SW-OPAs which was calculated as the average of $[dN(D_g)/d\log D_g]_{Aet}$ and $[dN(D_g)/d\log D_g]_{Nep}$.




## Appendix C: Truncation correction for total scattering

The nephelometer measurements were corrected for light scattered truncation for scattering angles (θ) between 0-7° and 170°-180° (Anderson et al., 1996; Anderson and Ogren, 1998). The truncation correction $C_{trunc}$ for total scattering was estimated as follows:

$$C_{trunc}\left(\lambda, m, N\left(D_g\right)\right) = \frac{\beta_{sca}\left(\lambda, m, N\left(D_g\right), 0° - 180°\right)}{\beta_{sca}\left(\lambda, m, N\left(D_g\right), 7° - 170°\right)} \tag{C1}$$

where $\beta_{sca}(\lambda, m, N(D_g), 0°-180°)$ is the scattering coefficient calculated for θ between 0° and 180°, and $\beta_{sca}(\lambda, m, N(D_g), 7°-170°)$ is the scattering coefficient calculated for θ between 7° and 170°. The Python PyMieScatt package (Sumlin et al., 2018) was used to retrieve $\beta_{sca}(\lambda, m, N(D_g), 0°-180°)$ and $\beta_{sca}(\lambda, m, N(D_g), 7°-170°)$ at a given λ by Mie calculation for homogeneous spheres by varying the particle size distribution $N(D_g)$ based on the complex refractive index. The SW-OPA geometrical size distribution $N_{sw-OPAs}(D_g)$ were used in the calculation.

For single homogeneous spherical particles, the scattering intensity function $|S(X_s, m, θ)|^2$ at given θ can be calculated using the Mie theory, where $X_s$ is the size parameter ($X_s = \pi \cdot D_g \cdot \lambda^{-1}$) and m is the complex refractive index of particles of geometrical diameter $D_g$. The scattering intensity function can be integrated over θ from 0° to 180° to obtain the single-particle total scattering efficiency (e.g., Anderson et al., 1996; Bohren and Hufmann, 1998):

$$Q_{sca}\left(\lambda, m, D_g\right) = \frac{1}{X_s^2} \cdot \int_0^\pi |S(X_s, m, θ)|^2 \cdot \sin θ \cdot dθ \tag{C2}$$

Finally, $\beta_{sca}$ can be calculated as the integral of $Q_{sca}$ over the particle size distribution $N(D_g)$ multiplied by the particle cross 580    sectional area (e.g., Anderson et al., 1996; Bohren and Hufmann, 1998):

$$\beta_{sca}\left(\lambda, m, N\left(D_g\right)\right) = \int_{D_{g,min}}^{D_{g,max}} \frac{\pi \cdot D_g^2}{4} \cdot Q_{sca}\left(\lambda, m, D_g\right) \cdot \frac{dN\left(D_g\right)}{d\log D_g} \cdot d\log D_g \tag{C3}$$

In this study, $\beta_{sca}(\lambda, m, N(D_g), 0°-180°)$ was computed at a given λ by the Mie_SD function for polydisperse size distributions in the Python PyMieScatt package by inputting the calculated $N_{sw-OPAs}(D_g)$, as the particle size distribution, and the corresponding complex refractive index. To determine $\beta_{sca}(\lambda, m, N(D_g), 7°-170°)$, first the angular scattering intensities $|S(X_s, m, θ)|^2$ for θ between 7° and 170° were computed by the ScatteringFunction angular function, and then used in 585    Eqs. (C2-3) to obtain $Q_{sca}$ and $\beta_{sca}$.



## Appendix D: Calculation of relevant parameters to correct the attenuation coefficient

This section provides a description of the calculation of the parameters in Eq. (5) which were used to retrieve the absorption coefficient $\beta_{abs}(\lambda)$ from the attenuation coefficient $\beta_{ATTN}(\lambda)$.

The parameter $\alpha(\lambda)$ was calculated using the formula from Arnott et al. (2005):

$$\alpha(\lambda) = A^{(d-1)} \cdot c \cdot \lambda^{-\text{å}_{sca} \cdot (d-1)} \tag{D1}$$

where A and $\text{å}_{sca}$ were obtained from the power-law fitting of $\beta_{sca}(\lambda)$ ($m^{-1}$) versus $\lambda$ (nm) weighted by the inverse of the variance of $\beta_{sca}(\lambda)$. The scattering coefficient $\beta_{sca}(\lambda)$ measured by the nephelometer, corrected for truncation, and extrapolated at the aethalometer $\lambda$ was used in the calculation. We assumed $c = 3.29 \cdot 10^{-4}$ and $d = 0.564$ as reported in Collaud Coen et al. (2010).

The obtained $\alpha(\lambda)$ increased with time as the particles were accumulating on the filter. Different $\lambda$ showed similar values of the $\alpha$ parameters. In the Icelandic dust samples, $\alpha(\lambda)$ ranged from around 0.005 to 0.02, which is within the range (0.002 to 0.02) reported by Di Biagio et al. (2019) for coarse mineral dust. The uncertainty on $\alpha(\lambda)$ varied between 17%-76% and was estimated by the error propagation through Eq. (D1) considering the uncertainty on the fitted parameters A and $\text{å}_{sca}$. The uncertainty on $\alpha(\lambda)$ was 35%-76% for the samples D3 and MIR45 which showed the largest errors on the fitted parameters,

29%-45% for the samples Maeli2 and H55, and 17%-27% for the sample Land1.

The multiple scattering effect correction $C_{ref}$ was extrapolated at 370, 470, 520, 590, 660, 880, and 950 nm by the linear regression of $C_{ref}$ at 450 and 660 nm estimated for mineral dust by Di Biagio et al. (2017). $C_{ref}$ varied from 4.3 at 370 nm to 3.3 at 950 nm. The uncertainty on $C_{ref}$ determined by Di Biagio et al. (2017) was 10%.

The loading effect correction $R(\lambda)$ was calculated from the spectral attenuation ATTN measured by the aethalometer using

the formula from Collaud Coen et al. (2010):

$$R(\lambda) = \left(\frac{1}{f(\lambda)} - 1\right) \cdot \frac{ATTN(\lambda)\%}{50\%} + 1 \tag{D2}$$

where $f(\lambda)$ depends on the aerosol absorption properties and is a function of $SSA(\lambda)$:

$$f(\lambda) = a\left(1 - SSA(\lambda)\right) + 1 \tag{D3}$$

We assumed $a = 0.74$ as reported in Collaud Coen et al. (2010). As an initial guess for $R(\lambda)$, the single scattering albedo in Eq. (D3) was estimated using $\beta_{sca}(\lambda)$ measured by the nephelometer, corrected for truncation, and extrapolated at the aethalometer $\lambda$, and $\beta_{abs}(\lambda)^*$ which is the absorption coefficient corrected only for the scattering effect (Di Biagio et al.,

610 2019):



$$\beta_{abs}(\lambda)^* = \beta_{ATTN}(\lambda) - \alpha(\lambda) \cdot \beta_{sca}(\lambda) \qquad \text{(D4)}$$

$$SSA(\lambda)^* = \frac{\beta_{sca}(\lambda)}{\beta_{sca}(\lambda) + \beta_{abs}(\lambda)^*} \qquad \text{(D5)}$$

$R(\lambda)$ increased with $\lambda$ but decreased with time as particles were loaded onto a new filter. $R(\lambda)$ obtained after the second iteration varied between 0.7 and 1 in different Icelandic dust samples which is within the range reported by Di Biagio et al. (2019) for mineral dust, from 0.5 to 1. The uncertainty on $R(\lambda)$ was also estimated by applying the error propagation method and was generally $\leq 10\%$ but varied between 10%-50% within the first 10 min that the particles were loaded onto a new filter.




## Open Research

Datasets for this research are included in this paper (and its supplementary information files). Example research codes are

available at the following GitHub repository: https://github.com/ClarissaBaldo/Research_codes_examples.

## Author contributions

CB, ZS, CDB, PF and JFD designed the experiments and discussed the results. ZS supervised the experimental and data analyses. CB performed the experiments at CESAM with support from ZS, CDB, PF, MC, EP, and JFD, and the data

analysis with contributions from ZS, ARM, GL, CS, DB, CDB, and PF. GL refined the research codes. The soil samples used for the experiments were collected by OA and PDW. CB prepared the manuscript with contributions from all co-authors.

## Acknowledgements

Clarissa Baldo is funded by the Natural Environment Research Council (NERC) CENTA studentship (grant no. NE/L002493/1). This paper is partly funded by the NERC highlight topic project (NE/S00579X/1). This project has received funding from the European Union's Horizon 2020 research and innovation programme through the EUROCHAMP-2020 Infrastructure Activity under grant agreement No 730997. Part of this work was supported by the COST Action inDust (CA16202) supported by COST (European Cooperation in Science and Technology). OA and PDW acknowledge financial

support from the Czech Science Foundation (project No. 20-06168Y). GL thanks the PhD studentships funded by China Scholarship Council. CNRS-INSU is gratefully acknowledged for supporting the CESAM chamber as a national facility as part of the French ACTRIS Research Infrastructure as well as the AERIS data center (www.aeris–data.fr) for distributing and curing the data produced by the CESAM chamber through the hosting of the EUROCHAMP datacenter (https://data.eurochamp.org).






**Table 1: Experiment-averaged single scattering albedo $SSA_{avg}(\lambda)$ ± estimated uncertainty at $\lambda$ = 370, 470, 520, 590, 660, 880, 950 nm of Icelandic dust for the base simulation. $SSA_{avg}(\lambda)$ results from the base simulation are consistent with the results from Test 1 and Test 2 (see Table S1).**

| Sample ID | $SSA_{avg}(\lambda)$ | | | | | | |
|---|---|---|---|---|---|---|---|
| | 370 nm | 470 nm | 520 nm | 590 nm | 660 nm | 880 nm | 950 nm |
| D3 | 0.93 ± 0.02 | 0.95 ± 0.01 | 0.96 ± 0.01 | 0.96 ± 0.01 | 0.96 ± 0.02 | 0.96 ± 0.01 | 0.96 ± 0.02 |
| H55 | 0.94 ± 0.06 | 0.96 ± 0.08 | 0.96 ± 0.08 | 0.96 ± 0.08 | 0.96 ± 0.08 | 0.96 ± 0.07 | 0.96 ± 0.07 |
| Land1 | 0.91 ± 0.05 | 0.94 ± 0.03 | 0.95 ± 0.03 | 0.95 ± 0.03 | 0.96 ± 0.04 | 0.96 ± 0.04 | 0.96 ± 0.04 |
| Maeli2 | 0.90 ± 0.03 | 0.93 ± 0.02 | 0.94 ± 0.02 | 0.95 ± 0.02 | 0.95 ± 0.01 | 0.96 ± 0.01 | 0.95 ± 0.02 |
| MIR45 | 0.90 ± 0.04 | 0.92 ± 0.03 | 0.93 ± 0.02 | 0.94 ± 0.03 | 0.94 ± 0.03 | 0.94 ± 0.03 | 0.94 ± 0.03 |


**Table 2: Experiment-averaged imaginary index $k_{avg}(\lambda)$ ± estimated uncertainty at $\lambda$ = 370, 470, 520, 590, 660, 880, 950 nm of Icelandic dust. $k_{avg}(\lambda)$ data are the mean of the results of the base simulation and Test 1 in Table S2.**

| Sample ID | $k_{avg}(\lambda)$ | | | | | | |
|---|---|---|---|---|---|---|---|
| | 370 nm | 470 nm | 520 nm | 590 nm | 660 nm | 880 nm | 950 nm |
| D3 | 0.004 ± 0.002 | 0.003 ± 0.001 | 0.002 ± 0.001 | 0.002 ± 0.001 | 0.002 ± 0 | 0.002 ± 0 | 0.002 ± 0 |
| H55 | 0.004 ± 0.001 | 0.004 ± 0.002 | 0.003 ± 0.001 | 0.002 ± 0.001 | 0.002 ± 0 | 0.002 ± 0.001 | 0.002 ± 0 |
| Land1 | 0.004 ± 0.001 | 0.004 ± 0.002 | 0.004 ± 0.002 | 0.003 ± 0.001 | 0.003 ± 0.001 | 0.002 ± 0.001 | 0.002 ± 0.001 |
| Maeli2 | 0.004 ± 0.002 | 0.004 ± 0.002 | 0.002 ± 0.001 | 0.003 ± 0.001 | 0.003 ± 0.001 | 0.002 ± 0.000 | 0.002 ± 0.001 |
| MIR45 | 0.004 ± 0.001 | 0.004 ± 0.001 | 0.003 ± 0.001 | 0.003 ± 0.001 | 0.003 ± 0.001 | 0.003 ± 0.000 | 0.003 ± 0 |



**Table 3: Experiment-averaged real index $n_{avg}(\lambda)$ ± estimated uncertainty at $\lambda$ = 370, 470, 520, 590, 660, 880, 950 nm of Icelandic**
**dust. $n_{avg}(\lambda)$ data are the mean of the results of the base simulation and Test 1 in Table S3.**

| Sample ID | $n_{avg}(\lambda)$ | | | | | | |
|---|---|---|---|---|---|---|---|
| | 370 nm | 470 nm | 520 nm | 590 nm | 660 nm | 880 nm | X950 nm |
| D3 | 1.60 ± 0.01 | 1.60 ± 0.01 | 1.60 ± 0.01 | 1.60 ± 0.01 | 1.60 ± 0.02 | 1.60 ± 0.02 | 1.60 ± 0.01 |
| H55 | 1.60 ± 0.01 | 1.60 ± 0.01 | 1.60 ± 0.01 | 1.60 ± 0.01 | 1.61 ± 0.02 | 1.60 ± 0.01 | 1.60 ± 0 |
| Land1 | 1.60 ± 0.01 | 1.60 ± 0.01 | 1.60 ± 0.01 | 1.60 ± 0.01 | 1.61 ± 0.01 | 1.60 ± 0.01 | 1.61 ± 0.01 |
| Maeli2 | 1.60 ± 0.01 | 1.60 ± 0.01 | 1.60 ± 0.02 | 1.61 ± 0.01 | 1.60 ± 0.01 | 1.61 ± 0.01 | 1.61 ± 0.01 |
| MIR45 | 1.60 ± 0.01 | 1.60 ± 0.01 | 1.60 ± 0.01 | 1.60 ± 0.01 | 1.61 ± 0.01 | 1.60 ± 0.01 | 1.60 ± 0.01 |



**Figure 1:** Schematic diagram of the method used to retrieve the shortwave (SW) single scattering albedo (SSA) and complex refractive index (m = n - ik) of Icelandic dust for λ from 370 to 950 nm. See main text for details. $\beta_{sca}(\lambda)$ is the spectral scattering coefficient retrieved from the nephelometer measurements, and θ is the light scattering angle. $\beta_{ATTN}(\lambda)$ and $\beta_{abs}(\lambda)$ are respectively the spectral attenuation and absorption coefficients retrieved from the aethalometer measurements. $dN(D_m)/dlogD_m$ is the particle number concentration measured by the SMPS – scanning mobility particle sizer, where $D_m$ is the electrical mobility diameter. $dN(D_{op})/dlogD_{op}$ is the particle number concentration measured by the GRIMM – optical particle counter, where $D_{op}$ is the optical-equivalent diameter. $dN(D_{g,SMPS-GRIMM})/dlogD_{g,SMPS-GRIMM}$ is the geometrical size distribution retrieved from the SMPS or GRIMM measurements, where $D_g$ is the geometrical diameter obtained from $D_m$ or $D_{op}$ using selected intervals for the dust particle dynamic shape factor (χ), and the real (n) and imaginary (k) part of the complex refractive index, respectively. The term SW-OPA (Shortwave Optical Properties Analyzers) refers to the aethalometer and nephelometer.






**Figure 2: Effective diameters $D_{eff}$ of dust particles sampled by the SW-OPAs and in CESAM from 30 min after the injection peak to 2.5 h. a-b) Base simulation; c-d) Test 1; e-f) Test 2. $D_{eff}$ was calculated for particles > 1 μm ($D_{eff,coarse}$) and ≤ 1 μm ($D_{eff,fine}$). Data were reported as 12-min average. In Test 1, corrections and calculations were performed using the SMPS and GRIMM data plus 1 SD uncertainty. In Test 2, we used the SMPS and GRIMM data minus 1 SD uncertainty (see section 2.2.1 in the main text for details). Sample IDs: D3 and Maeli2.**





**Figure 3: Extinction coefficient β<sub>ext</sub>(λ), absorption coefficient β<sub>abs</sub>(λ), and single scattering albedo SSA(λ) of Icelandic dust at λ = 370, 470, 520, 590, 660, 880, 950 nm, from 30 min after the injection peak to 2.5 h. The reported results as 12-min average refer to the base simulation.**






**Figure 4: Real index n(λ) and imaginary index k(λ) at λ = 370, 470, 520, 590, 660, 880, 950 nm, from 30 min after the injection peak to 2.5 h. a-b) Base simulation; c-d) Test 1; e-f) Test 2. In Test 1, corrections and calculations were performed using the SMPS and GRIMM data plus 1 SD uncertainty. In Test 2, we used the SMPS and GRIMM data minus 1 SD uncertainty (see section 2.2.1 in**
**the main text for details). Sample ID: Maeli2.**

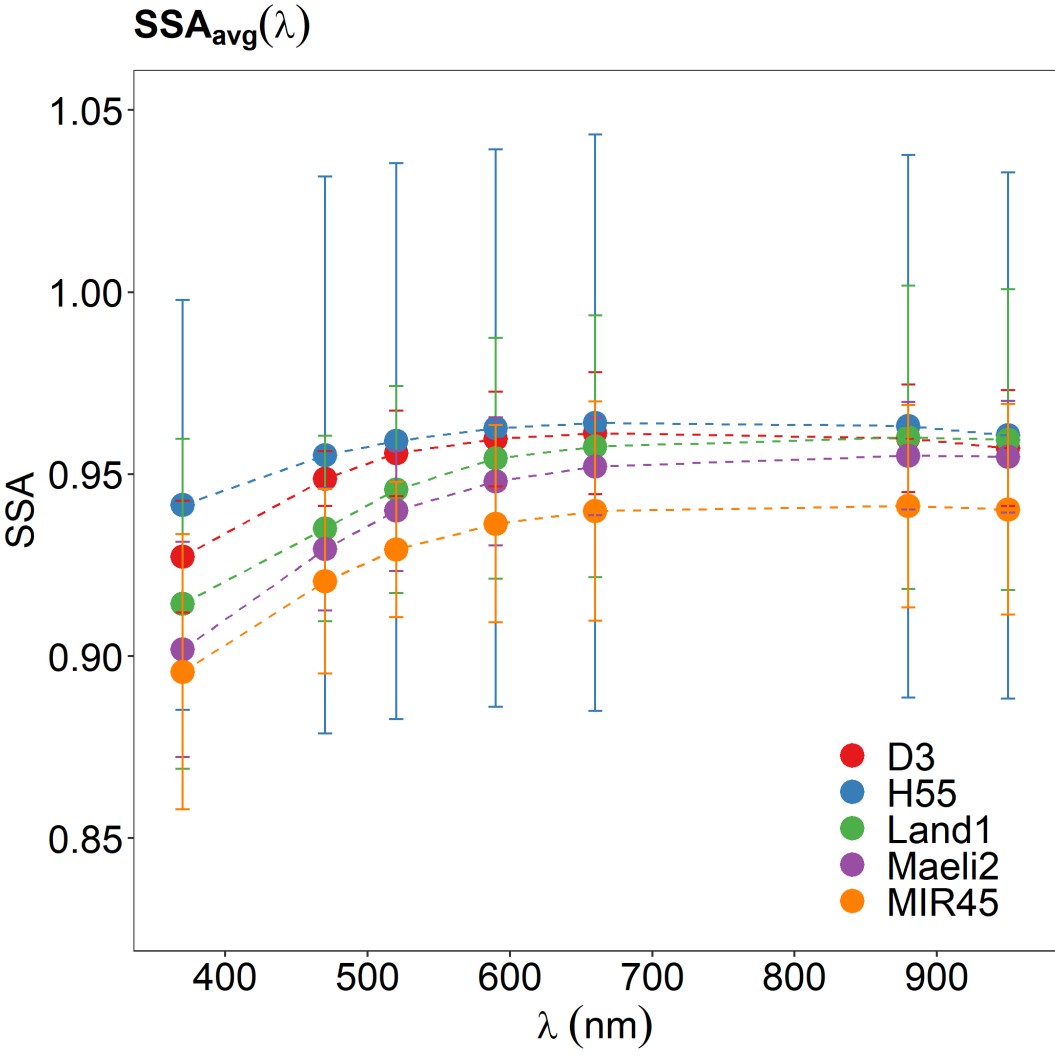

**Figure 5: Experiment-averaged single scattering albedo SSA$_{avg}$($\lambda$) at $\lambda$ = 370, 470, 520, 590, 660, 880, 950 nm of Icelandic dust samples for the base simulation (Table 1). SSA$_{avg}$($\lambda$) results from the base simulation are consistent with the results from Test 1 and Test 2 (see Table S1).**








**Figure 6: Comparison between the experiment-averaged imaginary index $k_{avg}(\lambda)$ at $\lambda$ = 370, 470, 520, 590, 660, 880, 950 nm of Icelandic dust (Table 2) and the initial estimates by Baldo et al. (2020) based on the mineralogical composition (low and high absorption case).**




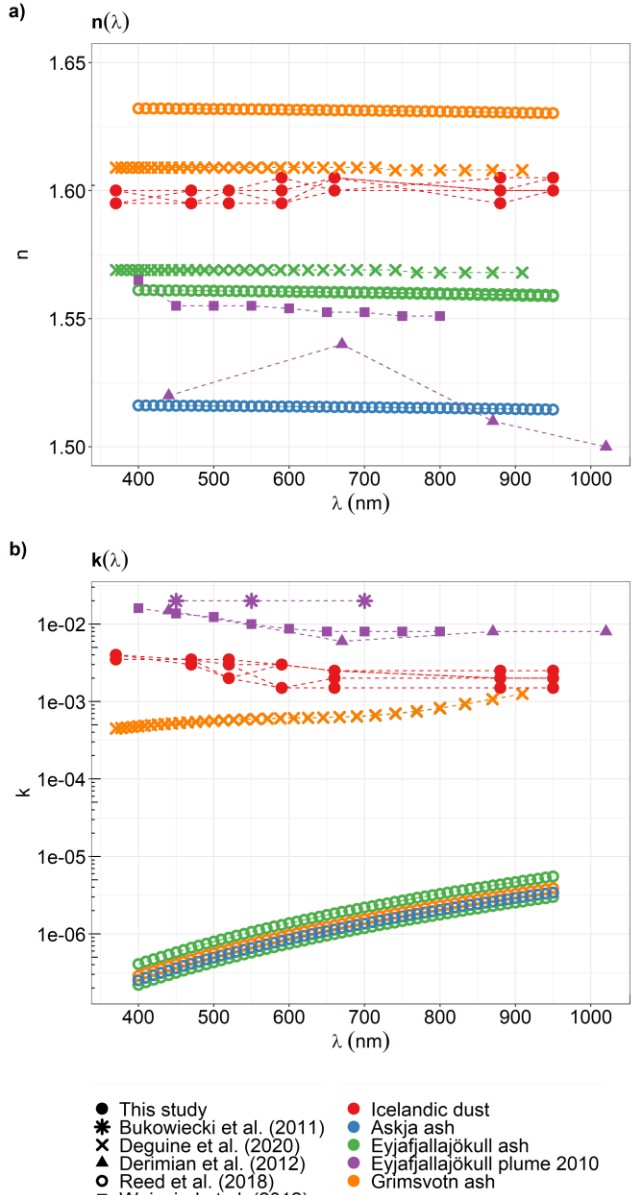

**Figure 7. Comparison between the experiment-averaged a) real index $n_{avg}(\lambda)$, and b) imaginary index $k_{avg}(\lambda)$ at $\lambda$ = 370, 470, 520, 590, 660, 880, 950 nm of Icelandic dust (Tables 2-3) and Icelandic volcanic ash (Bukowiecki et al., 2011; Deguine et al., 2020; Derimian et al., 2012; Reed et al., 2018; Weinzierl et al., 2012). Note that $n(\lambda)$ and $k(\lambda)$ data of Deguine et al. (2020) and Reed et al. (2018) were downloaded from http://eodg.atm.ox.ac.uk/ (last access 05 January 2023). Bukowiecki et al. (2011) reported an average k value of 0.02 between 450-700 nm, while n ranged from 1.60 to 1.80 (not shown).**




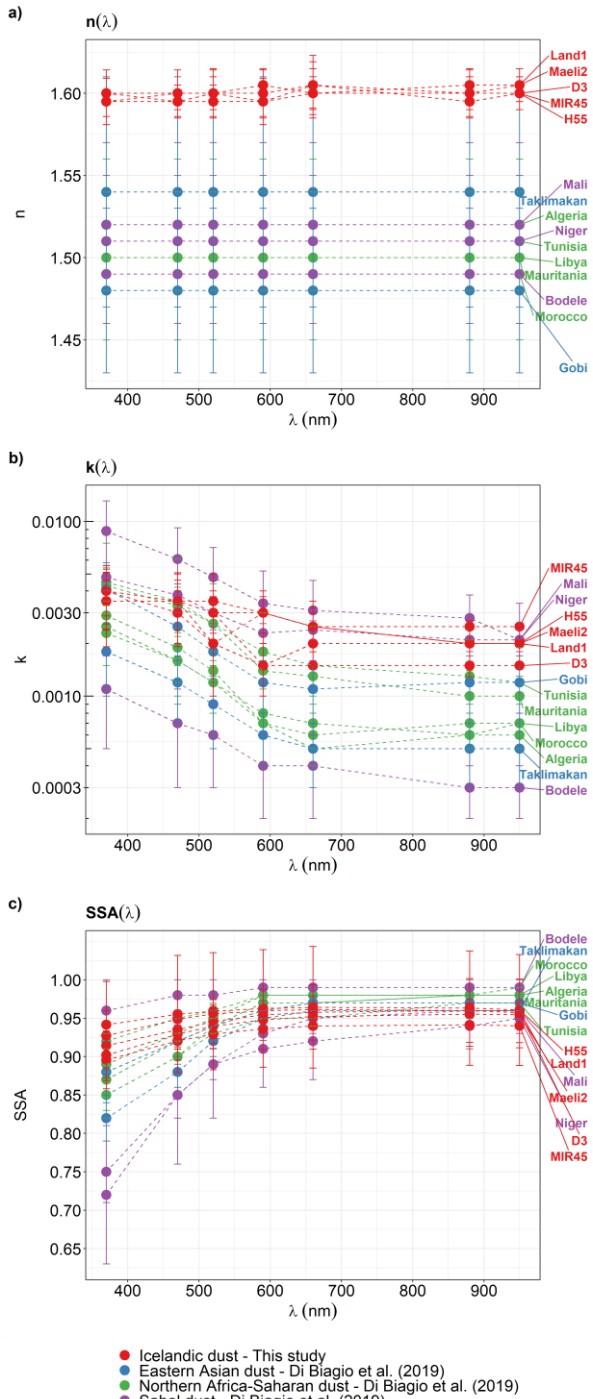

**Figure 8: Comparison between the experiment-averaged a) real index $n_{avg}(\lambda)$, b) imaginary index $k_{avg}(\lambda)$, and c) single scattering albedo $SSA_{avg}(\lambda)$ at $\lambda = 370, 470, 520, 590, 660, 880, 950$ nm of Icelandic dust (Tables 1-3) and mineral dust from major dust source regions at low-latitude (Di Biagio et al., 2019) including northern Africa-Sahara, the Sahel, and eastern Asia. Note we reported the real index of low-latitude mineral dust as the average value in the range 370-950 nm according to Table 4 of Di Biagio et al. (2019).**





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
