# Peer review of "Complex refractive index and single scattering albedo of Icelandic dust in the shortwave spectrum"

_EGUsphere, 2023_

## Referee Comment (RC1)

In this study, Baldo et al. retrieved the complex refractive index and the spectral single scattering albedo of Icelandic dust between 370-950 nm using measured dust size distribution as well as absorption and scattering coefficients from chamber experiments. The Icelandic dust samples were collected from five hotspots, the composition of which was examined in the companion paper. The optical properties of Icelandic dust were compared with previous studies on Icelandic dust, volcanic ash, and low-latitude dust. They found that the single scattering albedo and the real refractive index of Icelandic dust are within the range of low-latitude dust. However, the complex refractive index between 660-950nm is 2-8 times higher than most of the low-latitude dust, which is likely due to the high magnetite content in Icelandic dust. This finding suggests that Icelandic dust is more absorbing in the near-IR band and thus may have a more positive direct radiative effect.

High-latitude dust becomes increasingly important considering the land ice retreat under a warming climate. Icelandic dust is one of the key sources of high-latitude dust, while understanding of the radiative properties of Icelandic dust is limited. Therefore, this study is interesting and insightful. It fits well within the scope of ACP. I recommend the paper to be published after the following comments are addressed:

- This study uses suspended particles generated from natural parent soils. How are the suspended dust particles different from naturally emitted airborne dust particles, and how do the differences influence the estimates of Icelandic dust radiative properties?
- Section 2.1: I recommend including the location (latitude and longitude) and the time of collection for the five dust hotspots in the supplement. Although they are provided in the companion paper and other papers, it will be helpful for the reader to have the information easily available.
- Line 183-184: this sentence should be reworded slightly. Particle loss correction is further applied to the CESAM size distribution to determine the SW-OPAs size distribution. This is not clear to me until I read Appendix B.
- Section 2.3.2: it is not clear to me why the complex refractive index could be determined through the linear fit between measured and modeled $\beta$. Please add some explanation.
- The final k and n are determined by combining base simulation and Test 1. What is the uncertainty of ignoring results from Test 2? Will this cause k and n to be biased toward results from high particle number concentration (N+SD)?
- Line 448: what is the Fe content in low-latitude dust?
- Line 451: similar to the above comment, what is the magnetite fraction in low-latitude dust?
- The solar radiation peaks in the visible band (400 - 700 nm). The 660 - 950 nm wavelength band does not have intense solar radiation. How will this influence the significance of the results?

Editorial comments:

- Line 157: $dN(D_{g,GRIMM})/dlogD_{g,SMPS}$ should be $dN(D_{g,GRIMM})/dlogD_{g,GRIMM}$.
- Line 265: what is Eq. (6)(6) and Eq. (7)(6)?
- Line 396: the abbreviation of the Mýrdalssandur hotspot should also be provided here.
- Figure 1: I think all $B_{sca}$, $B_{abs}$, and $B_{ATTN}$ in the diagram should be $\beta_{sca}$, $\beta_{abs}$, and $\beta_{ATTN}$, respectively.
- Figure 3: please use a larger font for the title and the axis title of each panel. The $\beta_{sca}$ and $\beta_{abs}$ in the titles are particularly hard to read now.

- Figure S3: which $\chi$, n, and k are used here?

---

## Author Comment (AC1)

**Reviewer 1**

General Response: We thank Reviewer 1 for providing invaluable comments and insights. The comments have been addressed point by point below.

Please note, that the colours of all figures have been updated to meet the requirements for colour blindness as requested by the editor. Also note that the numbers of the tables and figures have been updated after implementing the reviewers' comments.

Page and Lines numbers in the responses refer to the manuscript tracked version.

**Comment: This study uses suspended particles generated from natural parent soils. How are the suspended dust particles different from naturally emitted airborne dust particles, and how do the differences influence the estimates of Icelandic dust radiative properties?**

Response: The method developed at CESAM (Di Biagio et al., 2017) allows parent soils to be used to generate dust aerosols with a realistic size distribution and mineralogy, including the main minerals responsible for absorption in the short-wavelength, SW, (Fe oxides) (e.g. Di Biagio et al., 2019) and LW, long-wavelength, (clay, quartz, calcite) (e.g., Di Biagio et al., 2017) spectra in concentrations typical of atmospheric conditions. The size distribution of particles sampled by the Shortwave Optical Properties Analyzers, SW-OPAs, is more representative of transported particles than of local aerosols near the source region, due to the loss of larger particles through the nephelometer and aethalometer sampling lines (Di Biagio et al., 2019; Di Biagio et al., 2017). In agreement with the size analysis of Di Biagio et al. (2017) and Di Biagio et al. (2019), the results of this study would be more suitable for understanding the radiative effects of Icelandic dust in regions affected by its long-range transport. Further analysis of the size distribution of Icelandic dust measured at CESAM and its evolution with respect to available field observations and with respect to low-latitude dust will be provided in a subsequent study (Di Biagio et al., in preparation).

The following paragraph was added to Section 2.2.1, Page 7, Lines 207-211:

"The size distribution of particles sampled by the SW-OPAs is more representative of transported particles than of local aerosols near the source region, due to the loss of larger particles through the nephelometer and aethalometer sampling lines (Di Biagio et al., 2019; Di Biagio et al., 2017). In agreement with the size analysis of Di Biagio et al. (2017) and Di Biagio et al. (2019), the results of this study would be more suitable for understanding the radiative effects of Icelandic dust in regions affected by its long-range transport."

**Comment: Section 2.1: I recommend including the location (latitude and longitude) and the time of collection for the five dust hotspots in the supplement. Although they are provided in the companion paper and other papers, it will be helpful for the reader to have the information easily available.**

Response: Table S1 with information on sampling sites has been added to the supporting information.

Furthermore, Section 2.1, Page 4, Line 127 has been updated as follows:

"Table S1 contains the information on sampling sites."

**Comment: Line 183-184: this sentence should be reworded slightly. Particle loss correction is further applied to the CESAM size distribution to determine the SW-OPAs size distribution. This is not clear to me until I read Appendix B.**

Response: Section 2.2.1, Page 7, Lines 198-207 have been updated based on the reviewer comment:

"The geometrical size distributions were first corrected for particle loss along the sampling lines of SMPS and GRIMM to obtain the size distributions of the particles suspended in CESAM. Subsequently, the particle loss correction was applied to the size distribution of particles within the chamber to determine the size distributions of particles sampled by the aethalometer and nephelometer. Furthermore, since the size distributions of the two instruments were similar (the percentage difference between the loss correction estimated for the aethalometer and that obtained for the nephelometer was < 11%), a common size distribution was calculated as the mean of the size distributions of the aethalometer and nephelometer. The resulting size distribution, referred to as $N_{sw-OPAs}(D_g)$, represents the size distribution detected by the Shortwave Optical Properties Analyzers (SW-OPAs)."

**Comment: Section 2.3.2: it is not clear to me why the complex refractive index could be determined through the linear fit between measured and modelled b. Please add some explanation.**

Response: We apologise for the confusion, we should not have used linear fit, the correct term is linear correlation.

Section 2.3.2, Page 11, Lines 311-321 have been updated as follows:

"In addition, the experiment-averaged spectral complex refractive index was determined based on the analysis of the linear correlation between $\beta_{abs,meas}(\lambda)$ vs $\beta_{abs,model}(\lambda)$, and $\beta_{sca,meas}(\lambda)$ vs $\beta_{sca,model}(\lambda)$ scenarios starting from 30 min after the dust injection peak to 2.5 h. To retrieve the experiment-averaged real index $n_{avg}(\lambda)$ and imaginary index $k_{avg}(\lambda)$, we updated the method applied to determine $k(\lambda)$ and $n(\lambda)$ at a 12-min resolution, as described in the preceding paragraph. For each $\beta_{abs,model}(\lambda)$ and $\beta_{sca,model}(\lambda)$ scenarios, we assumed that the input parameters $\chi$, $k(\lambda)$ and $n(\lambda)$ are constant throughout the experimental run. From the analysis of the linear correlation between $\beta_{abs,meas}(\lambda) \pm 1$ SD and $\beta_{abs,model}(\lambda)$ scenarios, and between $\beta_{sca,meas}(\lambda) \pm 1$ SD and $\beta_{sca,model}(\lambda)$ scenarios, we selected only the model estimates which showed a high correlation with observations ($R^2>0.70$). The modelled and measured spectral coefficients were then compared based on the root mean square error (RMSE) instead of using the %diff at individual time points. The uncertainty on $k_{avg}(\lambda)$ and $n_{avg}(\lambda)$ was estimated from the RSD of the k and n solutions. The uncertainty on $k_{avg}(\lambda)$ was up to 99%, while the uncertainty on $n_{avg}(\lambda)$ was < 2%."

**Comment: The final k and n are determined by combining base simulation and Test 1. What is the uncertainty of ignoring results from Test 2? Will this cause k and n to be biased toward results from high particle number concentration (N+SD)?**

Response: While the results obtained from averaging the baseline simulation and Test 1 may have a bias towards high particle number concentration, no significant difference was observed when the results of Test 2 were also included, for both the parameters n and k. Although the mean values of k tended to be higher when Test 2 was included in the average, particularly in the wavelength range of 660-950 nm, we calculated that the difference between the average with Test 2 included and the average without Test 2 is less than three times the square root of the sum of their squared uncertainties. Furthermore, it was decided to use the more conservative approach as the correlation between the measured and modelled $SSA_{avg}(\lambda)$ tends to be higher in the base simulation and Test 1 compared to Test 2 (Table S5), and the increase in $k_{avg}$ with $\lambda$ in Test 2 is hard to explain, suggesting that Test 2 results are not realistic.

Section 3.2, Page 14, Lines 404-415 have been updated as follows:

"Table S5 reports a summary of the comparison between $SSA_{avg}(\lambda)$ obtained from the measured and computed spectral coefficients for the base simulation, Test 1, and Test 2. Although the RMSE values were generally low, the correlation between the measured and modelled $SSA_{avg}(\lambda)$ tends to be higher in the base simulation and Test 1 compared to Test 2 (Table S5). Di Biagio et al. (2019) chose to average the k values from all three scenarios. Here the increase of $k_{avg}$ with $\lambda$ in Test 2 is hard to explain,

suggesting that Test 2 results are not realistic. Based on this, we chose to combine the results from the base simulation and Test 1 to obtain a single set of values for k(λ) (Table 2) and n(λ) (Table 3). While the results obtained from averaging the baseline simulation and Test 1 may have a bias towards high particle number concentration, no significant difference was observed when the results of Test 2 were also included, for both the parameters n and k. Although the mean values of k tended to be higher when Test 2 was included in the average, particularly in the wavelength range of 660-950 nm, the difference between the average with Test 2 included and the average without Test 2 is less than three times the square root of the sum of their squared uncertainties. For these reasons, it was decided to use the more conservative approach."

**Comment: Line 448: what is the Fe content in low-latitude dust?**

Response: The total iron content in typical low-latitude dust such us northern African and eastern Asian dust is around 1 wt%-8 wt% (e.g.Di Biagio et al., 2019; Jeong, 2008; Shi et al., 2011). which is lower than in Icelandic dust 11 wt%-13 wt% (Baldo et al., 2020).

Section 4.2, Page 17, Lines 494-496 have been updated as follows:

"The Icelandic dust showed a basaltic composition, with relatively high total Fe content, ranging from around 11 wt% in the sample H55 to 13 wt% in MIR45, compared to low-latitude dust (1 wt%-8 wt%; e.g. Di Biagio et al., 2019; Jeong, 2008; Shi et al., 2011)."

**Comment: Line 451: similar to the above comment, what is the magnetite fraction in low-latitude dust?**

Response: The magnetite fraction of typical low-latitude dust such us northern African and eastern Asian dust ranges from around 0.1 wt% - 0.8 wt%. The following paragraph from Baldo et al. (2020) provides a detailed comparison of the magnetite content between Icelandic dust and low-latitude dust:

Existing observations show that the magnetite content in African dust is generally below 0.1 wt % or not detectable (Lazaro et al., 2008; Moskowitz et al., 2016). Moskowitz et al. (2016) reported 0.6 wt % magnetite in surface sediments ($PM_{63}$) collected in proximity to the Tibesti volcanic system based on magnetic measurements. The content of magnetite reported in Asian dust source regions is in the range 0.1 wt %–0.8 wt % from magnetic measurements and XRD analysis (Jia et al., 2019; Maher et al., 2009; Song et al., 2014). In Icelandic dust, the magnetite content estimated from XRD measurements and sequential extractions is 1 wt %–2 wt %.

Section 4.2, Page 17, Lines 499-501 have been updated as follows:

"The magnetite fraction (1 wt%-2 wt %) was considerably higher than in low-latitude dust (generally < 0.1 wt%  or undetectable in northern African dust, and < 0.8 wt% in eastern Asian dust source regions; Jia et al., 2019; Lazaro et al., 2008; Maher et al., 2009; Moskowitz et al., 2016; Song et al., 2014) contributing to 7 %-15 % of total Fe."

**Comment: The solar radiation peaks in the visible band (400 - 700 nm). The 660 - 950 nm wavelength band does not have intense solar radiation. How will this influence the significance of the results?**

Response: Although the 700-1000 nm wavelength band is characterised by lower solar irradiance than the visible band, it still contributes approximatively 25% of the total spectral solar irradiance (e.g., Woods et al., 2009). Therefore, the increased absorption of Icelandic dust in this range may still influence the radiative balance and regional climate. The use of these new data in climate models will provide further insights into the radiative impact of Icelandic dust in the Arctic region.

The following paragraph was added to Section 5, Page 18, Lines 543-546:

"Although the 700-1000 nm wavelength band is characterised by lower solar irradiance than the visible band, it still contributes approximatively 25% of the total spectral solar irradiance (e.g., Woods et al., 2009). Therefore, the increased absorption of Icelandic dust in this range may still influence the radiative balance and regional climate."

**Editorial comments:**
- **Line 157: dN(Dg,GRIMM)/dlogDg,SMPS should be dN(Dg,GRIMM)/dlogDg,GRIMM.**
- **Line 265: what is Eq. (6)(6) and Eq. (7)(6)?**
- **Line 396: the abbreviation of the Mýrdalssandur hotspot should also be provided here.**
- **Figure 1: I think all Bsca, Babs, and BATTN in the diagram should be βsca, βabs, and βATTN, respectively.**
- **Figure 3: please use a larger font for the title and the axis title of each panel. The βsca and βabs in the titles are particularly hard to read now.**
- **Figure S3: which χ, n, and k are used here?**

Response: The editorial comments have been directly implemented in the text.

**Reviewer 2**

General Response: We thank Reviewer 2 for the positive feedback. The comments have been addressed point by point below.

Please note, that the colours of all figures have been updated to meet the requirements for colour blindness as requested by the editor. Also note that the numbers of the tables and figures have been updated after implementing the reviewers' comments.

Page and Lines numbers in the responses refer to the manuscript tracked version.

**Comment: Introduction: p2, line 45: Icelandic dust can reach several kilometres in altitude. Can you provide some numbers, 2km, 4 km, 8 km height. Some kind of a small summary. And also, how was it measured, ceilometer? aircraft? balloon? If dust at, e.g., 3 km height is observed, how can you be sure that this was Icelandic dust and not dust from other continents? So, I want to know a bit more about the range the Iceland dust can reach and potential horizontal transport scales.**

Response: The reviewer comment has been implemented in the text.

Section 1, Page 2, Lines 47-58 have been updated as follows:

"Icelandic dust can reach several kilometres in altitude, as shown in a study by Dagsson-Waldhauserova et al. (2019). They conducted six winter balloon launches in southern west Iceland during 2013-2016. Vertical profile measurements were collected using the Light Optical Aerosol Counter (LOAC) instrument, which provided aerosol size number distributions, concentrations, typologies, and basic meteorological parameters. LOAC estimates particle typologies for black carbon, mineral dust, volcanic dust, volcanic ash, ice particles, sea salt, and liquid particles, based on specific range referred to as 'specification zone' retrieved during laboratory experiments with different aerosols (Renard et al., 2016). The 'speciation zone' of Icelandic volcanic dust was established based on laboratory simulations using volcanic dust and ash samples collected in Iceland. Volcanic dust particles up to 10 μm were detected within 900 m altitude, those up to 5 μm mostly within 3.5 km altitude, while submicron particles were detected at altitudes up to 6 km. LOAC measurements were confirmed by in-situ ambient air quality monitoring stations of the Environmental Agency of Iceland for surface, CALIPSO and HYSPLIT model at different altitudes and MODIS true colour images with dust plumes, pointing to the local Icelandic dust sources during the Polar Vortex conditions above Iceland."

**Comment: Results: p11, line 302: I personally find these plots in Figure S2 very important for the study. It is always the question: Could the full size range be covered? And then these nice and important plots are at all in the supplementary.**

Response: The size ranges shown in Figure S2 are before the geometrical size distributions are corrected for particle loss along the instrument sampling systems. We estimated that the particles suspended in CESAM were in the size range up to 20 μm, while the particles sampled by the SW-OPAs (nephelometer and aethalometer) were in the size range up to 9 μm. Details of the particle loss correction can be found in Appendix B.

Figure S2 has been moved to the main text as current Figure 2. The following note was added to the figure caption:

"Note: The size ranges shown are before correcting the size distributions for particle loss along the instrument sampling systems. In the end, particles suspended in CESAM were up to 20 μm in size, while particles sampled by the SW-OPAs were up to 9 μm in size. Details of the particle loss correction can be found in Appendix B."

**Comment: Results: Figure 7: I was completely lost… All symbols (left column) in the legend are in black, and references are given (to each black symbol). In the figure, the symbols then have colors. In the legend, we have many colored full circles (but no references!). The k values for the volcanic ash are 3-4 orders of magnitude lower than for Icelandic dust. So, ash is totally non absorbing? Confusing?**

Response: We apologise for the confusion; the legend has been updated based on the reviewer comment to make the figure clearer (current Figure 8). The literature reports a wide range of values for the real (n) and imaginary (k) part of the complex refractive index in the SW-spectrum for Icelandic volcanic ash, which is discussed in detail in Section 4.1, Pages 15-16, Lines 435-458. Only the study by Reed et al. (2018) reported such low values for Icelandic volcanic ash; the updated figure should show this more clearly.

**Comment: Final point: Figure 8: This figure shows a nice comparison with other dusts from other major sources, but all measured by the same team! I miss a bit a comparison with other findings, by other groups, from different field campaigns (maybe from different deserts and continents). I remember, e.g., these SAMUM campaigns. There was this paper of Mueller et al, JGR 2010 (2009JD12520) with refractive index values etc. I could imagine, there are several other studies showing SSA and refractive index properties. Is it possible to have another figure or a table with different (independent) findings from different field studies… together with your findings?**

Response: The reviewer comment has been implemented in the text.

Figure S20 have been added to the supporting information.

Section 4.1, Page 16, Lines 459-463 have been updated as follows:

[revised manuscript text omitted]